# Theoretically Unmasking Inference Attacks Against LDP-Protected Clients in Federated Vision Models

Quan Nguyen [*1]   Minh N. Vu [*2]   Truc Nguyen [3]   My T. Thai [1]

## Abstract

Federated Learning enables collaborative learning among clients via a coordinating server while avoiding direct data sharing, offering a perceived solution to preserve privacy. However, recent studies on Membership Inference Attacks (MIAs) have challenged this notion, showing high success rates against unprotected training data. While local differential privacy (LDP) is widely regarded as a gold standard for privacy protection in data analysis, most studies on MIAs either neglect LDP or fail to provide theoretical guarantees for attack success rates against LDP-protected data.

To address this gap, we derive theoretical lower bounds for the success rates of low-polynomial-time MIAs that exploit vulnerabilities in fully connected or self-attention layers. We establish that even when data are protected by LDP, privacy risks persist, depending on the privacy budget. Practical evaluations on federated vision models confirm considerable privacy risks, revealing that the noise required to mitigate these attacks significantly degrades models' utility.

## 1. Introduction

Federated Learning (FL) (McMahan et al., 2017) is a decentralized machine learning paradigm where multiple devices or nodes (e.g., smartphones, edge devices, or distributed servers) collaboratively train a shared model while keeping their data localized. Due to this property, it has long been heralded as a robust solution for privacy-preserving machine learning. However, FL itself does not provide formal privacy guarantees, as the model updates (gradients

or weights) can still reveal sensitive information about the training data (Fang et al., 2024; Yang et al., 2023). A prominent attack against FL is the membership inference attack (MIAs) (Shokri et al., 2017; Hu et al., 2022b) in which the server seeks to determine whether a particular record was part of the model's training dataset.

To mitigate such privacy risks, local differential privacy (LDP) (Dwork et al., 2014; Wang et al., 2020) emerged as a prominent solution to limit the privacy leakage of local training data. Under LDP, individuals perturb their data locally before sharing it with the server, thereby eliminating the need for a trusted centralized curator. However, recent studies (Nguyen et al., 2023; Chen et al., 2021) demonstrated that tackling *active membership inference attacks (AMI)* conducted by dishonest FL servers requires adding large privacy-preserving noise that would also damage FL utility. In AMI, the server actively poisons the global model before distributing it to clients, enabling them to infer private information. Such attacks typically require multiple training iterations or the use of shadow models, resulting in non-trivial time complexity, and provide no theoretical privacy risks to FL. Building on this, (Vu et al., 2024) proposed a low-polynomial-time attack, presenting two active membership inference attacks on LLMs with guaranteed theoretical success rates on unprotected data.

Despite these advancements, most existing works rely heavily on empirical validation and lack a robust theoretical foundation or guarantees for attack success rates, especially under LDP. This challenge arises from the randomness of noise introduced by privacy mechanisms, which varies across iterations and clients, making it challenging to analyze the effectiveness of attacks in a theoretical framework.

This paper takes a step back and establishes a broader view of the principle of privacy risk imposed by dishonest servers in federated vision models under LDP from both theoretical and practical perspectives. Our study focuses on the active adversary setting, where the FL server acts dishonestly by manipulating the trainable weights of a vision model to breach privacy. Specifically, we aim to demonstrate that clients' data, even under LDP protection, are fundamentally vulnerable to AMI attacks carried out by dishonest servers. For that purpose, we analyze attacks that exploit the train-

---

[*]Equal contribution   [1]Department of CISE, University of Florida, USA. [2]Center for Nonlinear Studies, Los Alamos National Lab, USA [3]Computational Science Center, National Renewable Energy Laboratory, USA. Correspondence to: My T. Thai <mythai@cise.ufl.edu>.

*Proceedings of the 42nd International Conference on Machine Learning*, Vancouver, Canada. PMLR 267, 2025. Copyright 2025 by the author(s).

able fully connected (FC) layers and self-attention layers in FL updates as both are widely adopted in federated vision models. Our main contributions are summarized as follows:

- We derive theoretical lower and upper bounds (Theorem 1 and 2) on the success rates of a low-polynomial-time attack (Vu et al., 2024) that exploits vulnerabilities in FC layers, showing that privacy risks persist under LDP protection depending on the privacy budget.

- For transformer-based vision models such as ViTs (Dosovitskiy et al., 2021), we extend the attack on LLMs in (Vu et al., 2024) to continuous domain and derive theoretical lower bounds on the vulnerability of the self-attention mechanisms against a low-polynomial-time attack that exploit's the layer's memorization mechanism. (Theorem 3).

- Our experimental results in real-world state-of-the-art vision models such as ViTs and ResNet (He et al., 2016) demonstrate that AMI attacks achieve notably high success rates observed even under stringent LDP protection (i.e., small privacy budgets $\epsilon$) that considerably degrade the model's utility (Section 5). Furthermore, we consider both traditional FL where clients train a small model (ResNet) and the parameter-efficient-fine-tuning paradigm, where clients often utilize large foundation models for pre-training and fine-tune only some layers or parameters (ViTs).

## 2. Background and Related Works

**Federated Learning with Local Differential Privacy (FL-LDP).** In Federated Learning (FL) (McMahan et al., 2017), a central server orchestrates training while clients store data locally. The server initializes the model parameters $\theta$, and in each training iteration, a subset of clients computes gradients of the loss function $\mathcal{L}$ on local data $D$, i.e., $\dot{\theta} = \nabla_\theta \mathcal{L}_\Phi(D)$. These gradients are aggregated and sent to the server, which updates the parameters accordingly. Training continues until convergence.

To protect the privacy leakage in FL, Local Differential Privacy (LDP) (Dwork, 2006; Erlingsson et al., 2014) has been introduced. LDP is a privacy-preserving mechanism that mitigates risks by perturbing individual data before it leaves the client's device in FL. LDP ensures that the server cannot infer sensitive client information directly from the shared data as defined below.

**Definition 1.** $\varepsilon$-LDP. A randomized mechanism $\mathcal{M}$ satisfies $\varepsilon$-LDP if, for any two inputs $x$ and $x'$ and all possible outputs $\mathcal{O} \in \text{Range}(\mathcal{M})$,

$$Pr[\mathcal{M}(x) = \mathcal{O}] \le e^\varepsilon Pr[\mathcal{M}(x') = \mathcal{O}],$$

where $\varepsilon$ is the privacy budget, and Range($\mathcal{M}$) denotes all possible outputs of $\mathcal{M}$. In FL-LDP, $\varepsilon$ controls the privacy-utility trade-off: smaller $\varepsilon$ enhances privacy by introducing more noise to the data, but this can reduce the utility of the aggregated updates for the global model. While there are other privacy-preserving techniques for FL, such as secure aggregation using multi-party computation (SMPC) or homomorphic encryption (Bonawitz et al., 2017; Nguyen et al., 2023), these are orthogonal research directions and are discussed separately in Appendix. I.

**Federated Foundation Models via Parameter-Efficient Fine-Tuning (PEFT).** PEFT methods adapt large pre-trained models to specific tasks by modifying a small subset of parameters, reducing computational and storage costs. Key approaches include LoRA (Hu et al., 2022a), which adds trainable low-rank matrices; Adapter Modules (Yin et al., 2023), which insert lightweight layers; BitFit (Zaken et al., 2022), which updates bias terms; and Prompt Tuning (Lester et al., 2021), which optimizes input embeddings. In our work, we specifically analyze scenarios where trainable layers are fully connected or self-attention layers.

**Membership Inference Attacks (MIAs) in FL.** MIAs in FL aim to identify if a specific data point was part of a client's training set. Although FL keeps data local, model updates exchanged between clients and the server can still leak information. Passive attacks (Shokri et al., 2017; Zhang et al., 2020) involve an honest-but-curious server observing the model updates, while **Active Membership Inference** (AMI) attacks involve a dishonest server poisoning the global models, e.g., maliciously modifying model parameters, before dispatching them to clients. The first AMI attack in FL was introduced by (Nasr et al., 2019), relying on multiple FL iterations. A stronger, single-iteration AMI attack requiring training a separate neural network was later proposed by (Nguyen et al., 2023). Both approaches have non-trivial time complexity and do not establish theoretical privacy risks in FL. Recently, (Vu et al., 2024) introduced two AMI attacks that exploit fully connected and attention layers in LLMs, achieving a high success rate in compromising membership information of unprotected client data.

The primary focus of our study is to demonstrate the existence of low-complexity adversaries with provably high attack success rates, particularly when the data is protected by any ideal LDP mechanism. Since our research aims to assess the resilience of privacy-preserving techniques against real-world adversarial threats, we chose to examine two state-of-the-art AMI attacks on the FC and Attention layers proposed by (Vu et al., 2024), which have low-polynomial time complexity. These attacks allow the server to exploit FC layers to perfectly infer membership information (Theorem 1 in (Vu et al., 2024)) and to exploit self-attention layers to achieve a similarly high success rate (Theorem 2

in (Vu et al., 2024)). However, their theoretical analysis is only applicable to the non-LDP setting.

# 3. AMI Attacks

## 3.1. AMI threat models

The AMI threat models under LDP are formalized through the security games $\mathsf{Exp}_{\mathsf{LDP}}^{\mathsf{AMI}}$, as described in Fig. 1, following standard security frameworks (Nguyen et al., 2023; Vu et al., 2024). Further details about the security games can be found in Appendix B. In $\mathsf{Exp}_{\mathsf{LDP}}^{\mathsf{AMI}}$, the adversarial server $\mathcal{A}^{\mathcal{D}}$ (superscript $\mathcal{D}$ indicates that the server knows the data distribution of the client's private data) comprises three components: $\mathcal{A}_{\mathsf{INIT}}^{\mathcal{D}}$, $\mathcal{A}_{\mathsf{ATTACK}}^{\mathcal{D}}$, and $\mathcal{A}_{\mathsf{GUESS}}^{\mathcal{D}}$. In $\mathsf{Exp}_{\mathsf{LDP}}^{\mathsf{AMI}}$, a random bit $b$ determines if a target sample $T$ is in the client's data $D$. Each client applies LDP to perturb their data, generating $D' = \mathcal{M}^{\varepsilon}(D) = \{\mathcal{M}^{\varepsilon}(X)\}_{X \in D}$. The server's $\mathcal{A}_{\mathsf{INIT}}^{\mathcal{D}}$ selects a model $\Phi$, and $\mathcal{A}_{\mathsf{ATTACK}}^{\mathcal{D}}$ crafts parameters $\theta$ using $T$. Clients compute gradients $\dot{\theta} = \nabla_{\theta} \mathcal{L}_{\Phi}(D')$ and send them back. With $\dot{\theta}$, $\mathcal{A}_{\mathsf{GUESS}}^{\mathcal{D}}$ infers $b$, effectively identifying whether $T \in D$. The advantage of the adversarial server $\mathcal{A}^{\mathcal{D}}$ in the security game is given by:

$$\mathbf{Adv}_{\mathsf{LDP}}^{\mathsf{AMI}}(\mathcal{A}^{\mathcal{D}}) = 2\Pr[\mathsf{Exp}_{\mathsf{LDP}}^{\mathsf{AMI}}(\mathcal{A}^{\mathcal{D}}) = 1] - 1 \quad (1)$$
$$= \Pr[b' = 1 | b = 1] + \Pr[b' = 0 | b = 0] - 1$$

where $\frac{1}{2}\Pr[b' = 1 | b = 1] + \frac{1}{2}\Pr[b' = 0 | b = 0]$ denotes the success rate of the attack. The existence of an adversary with a high advantage implies a high privacy risk/vulnerability of the protocol described in the security game.

## 3.2. FC-based AMI adversary

We analyze the FC-based AMI adversary first introduced in (Vu et al., 2024). In that paper, they proved the existence of an AMI adversary that exploits two FC layers to achieve a perfect membership inference success rate for unprotected data. The FC-based adversary $\mathcal{A}_{\mathsf{FC}}^{\mathcal{D}}$ is designed to detect a target sample $T$ with dimension $d_T$ within local training data $D$ during FL training. For analysis of FC-based adversary, we represent the dataset as $D = \{X_i\}_{i=1}^{n}$, where $X_i \in \mathcal{X}$, and $\mathcal{X} \subseteq \mathbb{R}^{d_X}$. The model $\Phi$ introduces two adversarial fully connected (FC) layers. The first layer has weights $W_1$ of size $2d_T \times d_T$ and biases $b_1$ of size $2d_T$, structured to encode the target $T$. The second layer outputs a single neuron, with weights $W_2[1, :]$ and bias $b_2[1]$ set to target the presence of $T$. These parameters are defined as:

$$W_1 \leftarrow \begin{bmatrix} I_{d_T} \\ -I_{d_T} \end{bmatrix}, \quad b_1 \leftarrow \begin{bmatrix} -T \\ T \end{bmatrix} \quad (2)$$
$$W_2[1, :] \leftarrow -1_{d_T}^{\top}, \quad b_2[1] \leftarrow \tau^{\mathcal{D}} \quad (3)$$

where $\tau^{\mathcal{D}}$ controls the allowable $L_1$-distance between inputs and $T$. Upon receiving input $X$, the two FC layers compute

$z_0 := \max\{b_2[1] - \|X - T\|_{L_1}, 0\}$. If $X = T$, $z_0$ activates, and the gradient of $b_2[1]$ is non-zero. For $b_2[1] = \tau^{\mathcal{D}} > 0$ small enough, $z_0 = 0$ for $X \neq T$, leaving the gradient zero. The adversary uses the gradient of $b_2[1]$ as an indicator of the presence of $T$ in the local data. A non-zero gradient implies $T$ exists, while a zero gradient indicates it does not. The FC attack $\mathcal{A}_{\mathsf{FC}}^{\mathcal{D}}$ does not need any distributional information to work on unprotected data. In fact, the attacker just needs to specify $\tau^{\mathcal{D}}$ (2) small enough such that $\tau^{\mathcal{D}} < \|X_1 - X_2\|_{L_1}$ for any $X_1 \neq X_2$ in the model's dictionary. Since the dictionary or the pre-trained feature extractor is public, selecting $\tau^{\mathcal{D}}$ does not require any additional information. The description of $\mathcal{A}_{\mathsf{FC}}^{\mathcal{D}}$ is given in Appendix C.1.

## 3.3. Attention-based AMI adversary

The proposed $\mathcal{A}_{\mathsf{Attn}}^{\mathcal{D}}$ (Vu et al., 2024) leverages the memorization capability of self-attention, a property that was indirectly explored in (Ramsauer et al., 2021). That study demonstrates that self-attention can be interpreted as equivalent to the *Hopfield* layer, which is specifically designed to integrate memorization directly within the layer. Building on this perspective, $\mathcal{A}_{\mathsf{Attn}}^{\mathcal{D}}$ employs a tailored configuration of attention to facilitate the memorization of local training data while selectively excluding the target of inference.

The dataset is represented as $D = \{x_i\}_{i=1}^{n}$, where $x_i \in \mathcal{X}$, $\mathcal{X} \subseteq \mathbb{R}^{d_x \times N_x}$, and each column $x_j \in \mathbb{R}^{d_x}$ is referred to as a *pattern*. Since $\mathcal{A}_{\mathsf{Attn}}^{\mathcal{D}}$ operates at a pattern level, the target of inference is the pattern derived from embedding the target $T$, denoted as $v \in \mathbb{R}^{d_x}$. The underlying intuition of this approach involves configuring an attention head to memorize the input batch while excluding the target pattern. This configuration introduces a measurable discrepancy between the output of the filtered attention head and that of a non-filtered head. The resulting gap can then be exploited to infer the victim's data. A detailed explanation of the components of this attack are further elaborated in Appx. C.2 or readers can refer to (Vu et al., 2024) for the full description of the attack.

## 3.4. Attention-based AMI attack against Vision Transformer

While (Vu et al., 2024) focuses on exploiting the attention layer in LLMs to infer the presence of a pattern in a dataset, we further extend this attack to the continuous image domain, where we exploit the attention layer used in ViTs. We formulate the attack $\mathcal{A}^{\mathcal{D}}_{\mathsf{Attn}}$ in the context of attacking ViT models as follow: Given an image $I \in \mathbb{R}^{H \times W \times C}$, the image is divided into $L$ non-overlapping patches (see Fig. 14). The patches are flattened into vectors, and projected into an embedding space using a linear projection matrix $W_{\mathsf{embed}}$. The result of this embedding layer is:

$$x_j = \mathsf{Flatten}(I_j)W_{\mathsf{embed}} + p_j, \quad j = 1, \ldots, L$$

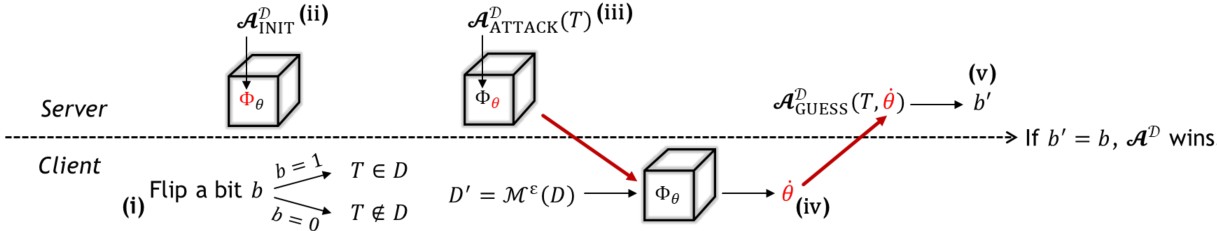

*Figure 1.* Active inference security game under LDP: a random bit $b$ determines the state of the data $D$ **(i)**, the server $\mathcal{A}^{\mathcal{D}}$ specifies $\Phi$ **(ii)** and $\theta$ **(iii)**, gradients on LDP-protected data $D'$ are sent back **(iv)**, and $\mathcal{A}^{\mathcal{D}}$ guesses $b$ **(v)**.

where $I_j \in \mathbb{R}^{\frac{H}{\sqrt{L}} \times \frac{W}{\sqrt{L}} \times C}$ represents the $j$-th patch of the image $I$ and $p_j$ is the corresponding positional encoding. The resulting set of embeddings $\{x_j\}_{j=1}^{L}$ is then passed through the Vision Transformer (ViT) architecture, where the attention mechanism operates on these embeddings. When incorporating LDP noise into ViTs, we apply the noise directly to these embeddings before they enter the attention layers. We employ a similar attack strategy to the one used in the attention-based AMI adversary, but in the context of the ViT's embeddings. Implementation details of the attack are given in Appx. G.2.

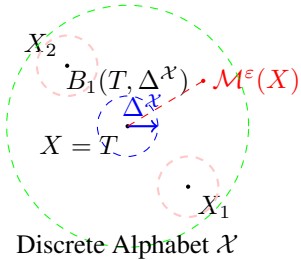

(a) The protected version of the target $\mathcal{M}^\varepsilon(X)$ jumps out of $B_1(T, \Delta^{\mathcal{X}})$.

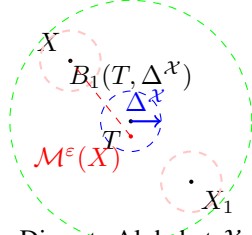

(b) $X \neq T$ such that $\mathcal{M}^\varepsilon(X) \in B_1(T, \Delta^{\mathcal{X}})$

*Figure 2.* Scenarios when $\mathcal{A}_{\mathsf{FC}}^{\mathcal{D}}$ fail.

# 4. Privacy Leakage Analysis

This section presents our theoretical analysis for assessing the risk of leaking membership information of users' local training data in FL under LDP. Given the security game

$\mathsf{Exp}_{\mathsf{LDP}}^{\mathsf{AMI}}$ defined in Section 3.1, we generalize the lower bound and upper bound for the advantage of the adversarial server $\mathcal{A}_{\mathsf{FC}}^{\mathcal{D}}$ in Theorem 1 and Theorem 2, respectively. Finally, we provide the lower bound for the advantage of $\mathcal{A}_{\mathsf{Attn}}^{\mathcal{D}}$ under LDP in Theorem 3.

## 4.1. FC-based AMI on LDP-Protected Data

We now theoretically show that the adversary $\mathcal{A}_{\mathsf{FC}}^{\mathcal{D}}$ constructed in Subsect. 3.2 can also be used to expose the true privacy risks of LDP-protected data w.r.t AMI in FL and state it in Theorem 1. First, we need to lay out some assumptions on the data and the LDP mechanisms.

The data $X$ is assumed to be from a discrete alphabet $\mathcal{X}$, in which the $L_1$ metric is well-defined. In our analysis, we denote $\mathcal{X}$ as the set of possible output values of the LDP algorithm (see remark 2). We denote $\Delta^{\mathcal{X}} := \min_{X,Y \in \mathcal{X}} \|X - Y\|_{L_1}/2$. Note that $\Delta^{\mathcal{X}}$ is a statistic of $\mathcal{D}$ and is known by the server. We denote $B_1(X, \Delta^{\mathcal{X}})$ to be a ball of radius $\Delta^{\mathcal{X}}$ centering around $X$ in the $L_1$ norm. Given an LDP mechanism $\mathcal{M}$ with budget $\varepsilon$ applied on an alphabet $\mathcal{X}$, $P_{\mathcal{M}^\varepsilon}$ denotes the probability that the protected version of a point is not in the ball of radius $\Delta^{\mathcal{X}}$ centering at that point: $P_{\mathcal{M}^\varepsilon} := \Pr\left[\mathcal{M}^\varepsilon(X) \notin B_1(X, \Delta^{\mathcal{X}})\right]$. Intuitively, a smaller $\varepsilon$ would impose more LDP noise, resulting in a larger $P_{\mathcal{M}^\varepsilon}$.

**Theorem 1.** *Given the security game* $\mathsf{Exp}_{\mathsf{LDP}}^{\mathsf{AMI}}$, *there exists an AMI adversary* $\mathcal{A}_{\mathsf{FC}}^{\mathcal{D}}$ *whose time complexity is* $\mathcal{O}(d_X^2)$ *such that* $\mathbf{Adv}_{\mathsf{LDP}}^{\mathsf{AMI}}(\mathcal{A}_{\mathsf{FC}}^{\mathcal{D}}) \geq 1 - \frac{n+|\mathcal{X}|-1}{|\mathcal{X}|-1} P_{\mathcal{M}^\varepsilon}$, *where $n$ is the size of the dataset $D$, $|\mathcal{X}|$ is the cardinality of the possible output values of the LDP-mechanism and $P_{\mathcal{M}^\varepsilon}$ is the probability that the LDP-mechanism makes the protected version of data point inside the neighborhood of another data point. (Proof in Appx. D.1)*

We use $\mathcal{A}_{\mathsf{FC}}^{\mathcal{D}}$ constructed in Subsect. 3.2 with $\tau^{\mathcal{D}} = \Delta^{\mathcal{X}}$ to show Theorem 1. Given input $X$, target $T$ and LDP mechanism $\mathcal{M}^\varepsilon$, the goal of $\mathcal{A}_{\mathsf{FC}}^{\mathcal{D}}$ is to configure the first two FC layers so that the first row of the second layer computes $z_0 := \max\{\tau^{\mathcal{D}} - \|\mathcal{M}^\varepsilon(X) - T\|_{L_1}, 0\}$. If $\|\mathcal{M}^\varepsilon(X) - T\|_{L_1} < \tau^{\mathcal{D}}$, then $z_0$ activates and the gradient of $b_2[1]$ is

non-zero. For $\tau^{\mathcal{D}} = \Delta^{\mathcal{X}}$, if $\mathcal{M}^{\varepsilon}(X) \notin B_1(T, \Delta^{\mathcal{X}})$, then $z_0 = 0$, leaving the gradient zero. Conversely, if $\mathcal{M}^{\varepsilon}(X) \in B_1(T, \Delta^{\mathcal{X}})$, then $z_0 > 0$ and the gradient $\dot{\theta}(b_2[1]) > 0$. The adversary uses the gradient of $b_2[1]$ as an indicator of the presence of $T$ in the local data. A non-zero gradient implies $T$ exists in the data, while a zero gradient indicates it does not. Intuitively, $\mathcal{A}_{\mathsf{FC}}^{\mathcal{D}}$ fails if either (i) the protected version of the data $\mathcal{M}^{\varepsilon}(X)$ jumps out of $B_1(T, \Delta^{\mathcal{X}})$ when $X = T$ or (ii) there is an $X \neq T$ such that $\mathcal{M}^{\varepsilon}(X) \in B_1(T, \Delta^{\mathcal{X}})$. These scenarios are illustrated in Fig. 2. The probabilities of the two events are bounded by $P_{\mathcal{M}^{\varepsilon}}$ and $nP_{\mathcal{M}^{\varepsilon}}/(|\mathcal{X}| - 1)$, respectively (see Appx. D.1). Theorem 1 demonstrates the trade-off between privacy and data utility: a highly protected data would have high $P_{\mathcal{M}^{\varepsilon}}$, thus lowering the advantage of the adversary; however, its distortion from the original data is large as a result.

**Remark 1. Lower bound of Theorem 1.** It is non-trivial to obtain $P_{\mathcal{M}^{\varepsilon}}$ for Theorem 1 due to the dependencies on the data as well as the specific LDP mechanisms. To demonstrate the intuition behind the proof, we provide an example of how to derive $P_{\mathcal{M}^{\varepsilon}}$ for Generalized Random Response (GRR (Warner, 1965)), a classical LDP algorithm, and the corresponding theoretical lower bound for GRR-protected data in Theorem 4 (Appx. E). We also simulate the lower theoretical bound in Theorem 1 for data protected by LDP algorithms BitRand (Jiang et al., 2022), GRR, dBitFlipPM (Ding et al., 2017) and RAPPOR (Erlingsson et al., 2014) in Figs. 7 and 8. Those theoretical lower bounds are shown along with the success rates of some AMI attacks (discussed in Section 3) for comparison.

**Remark 2. Cardinality of $\mathcal{X}$.** $|\mathcal{X}|$ is dependent on the specific LDP algorithm used. For binary LDP algorithms such as Binary Randomized Response (Warner, 1965) or RAPPOR (Erlingsson et al., 2014), $|\mathcal{X}| = 2$ due to the binary nature of the values being perturbed (e.g., a "yes/no" or "0/1" response). In contrast, for generalized $k$-ary randomized response algorithms, such as Generalized Randomized Response (GRR) or k-RAPPOR, $|\mathcal{X}| = k$, where $k > 2$ represents the cardinality of the set of possible output values of the algorithm. This allows for more nuanced privacy-preserving mechanisms, where the response set consists of $k$ different values, each with a specific probability distribution determined by the privacy parameters of the algorithm. In modern bit-flipping algorithms such as OME or BitRand, the original data or embedding features are first converted into binary vectors of size $b$. The LDP mechanisms are then applied on top of those binary representations of the signal by flipping random bits. In this case, $|\mathcal{X}| = 2^b$. For large enough $b$, $\frac{n + |\mathcal{X}| - 1}{|\mathcal{X}| - 1} \approx 1$ and $\mathbf{Adv}_{\mathsf{LDP}}^{\mathsf{AMI}}(\mathcal{A}_{\mathsf{FC}}^{\mathcal{D}}) \approx 1 - P_{\mathcal{M}^{\varepsilon}}$.

**Theorem 2.** *For all AMI adversary $\mathcal{A}$ of the security game* $\mathsf{Exp}_{\mathsf{LDP}}^{\mathsf{AMI}}$*, we have*

$$\mathbf{Adv}_{\mathsf{LDP}}^{\mathsf{AMI}}(\mathcal{A}_{\mathsf{FC}}^{\mathcal{D}}) \leq \frac{e^{\epsilon} - 1}{e^{\epsilon} + 1} \tag{4}$$

*(Proof in Appx. F)*

This theorem shows the theoretical upper bound for $\mathbf{Adv}_{\mathsf{LDP}}^{\mathsf{AMI}}(\mathcal{A}_{\mathsf{FC}}^{\mathcal{D}})$, given the privacy budget $\epsilon$. Accordingly, we measure the adversary's attack success rate as $\frac{1}{2}(1 + \mathbf{Adv}_{\mathsf{LDP}}^{\mathsf{AMI}}(\mathcal{A}_{\mathsf{FC}}^{\mathcal{D}}))$ (see Equation 1) and plot the theoretical upper bound alongside the lower bound of AMI attack success rate and empirical success rate against real-world datasets under LDP protection in Figs. 7 and 8.

### 4.2. Attention-based AMI on LDP-Protected Data

Our theoretical result on the vulnerability of private data to Attention-based AMI attack under LDP is quantified on *Separation of Patterns*, an intrinsic measure of data:

**Definition 2.** (Separation of Patterns (Ramsauer et al., 2021)). For a pattern $x_i$ in a data point $X = \{x_j\}_{j=1}^{N_X}$, its separation $\Delta_i$ from $X$ is $\Delta_i := \min_{j, j \neq i} \left( x_i^{\top} x_i - x_i^{\top} x_j \right) = x_i^{\top} x_i - \max_j x_i^{\top} x_j$. We say the pattern $i$ is separated from the data point $X$ if $\Delta_i > 0$. We say $X$ is $\Delta$-separated if $\Delta_i \geq \Delta$ for all $i \in \{1, \cdots, N_X\}$. A data $D$ is $\Delta$-separated if all $X$ in $D$ are $\Delta$-separated.

For the attention-based attack, we represent the victim's dataset as $D = \{X_i\}_{i=1}^{n}$, where $X_i \in \mathcal{X}$, and $\mathcal{X} \subseteq \mathbb{R}^{d_X \times N_X}$. For any 2-dimensional array $X$, each column $x_j \in \mathbb{R}^{d_X}$ is referred to as a *pattern*. Since the LDP mechanisms are generally applied at a pattern level in 2-dimensional data (Qu et al., 2021; Yue et al., 2021), the distortion imposed by LDP is modeled by a noise $r_i$ added to each pattern: $X^{\varepsilon} = \mathcal{M}^{\varepsilon}(X) = \{x_i + r_i\}_{i=1}^{N_X}$. We assume $r_i$ is bounded by a norm budget $R^{\varepsilon}$ that is defined by specific mechanisms and applications. $M$ denotes the maximum $L_2$-norm of all patterns, defined as $M = \max_{X \in D} \max_{x_j \in X} \|x_j\|$. The adjusted pattern's norm is then upper-bounded by $M^{\varepsilon} = \sqrt{M^2 + R^{\varepsilon 2}}$. Regarding data separation under LDP, denoted by $\Delta^{\varepsilon}$, whose value is not easily obtainable even when the LDP mechanism is known, we can generally expect $\Delta^{\varepsilon} \geq \Delta$. The reason is, as the noise $r_i$ is independent of the patterns, it makes the patterns less aligned. This intuition is demonstrated via an example in Appx. D.3.

The notion of $\Delta^{\varepsilon}$-separated helps capture the intrinsic difficulty of the data for the inference task: the less separating the data, i.e., a smaller $\Delta^{\varepsilon}$, the harder for the adversary to detect the patterns. However, it is not beneficial to impose a low separation on the data in practice since it would impair the model's performance. Note that, if $D$ is considered as the data after preprocessing, $\Delta^{\varepsilon}$ can be manipulated by the choice of preprocessing methods for the FL model, which are often specified by the server. We are now ready to state Theorem 3 that analyzes the vulnerability of LDP-protected data to Attention-based AMI in FL.

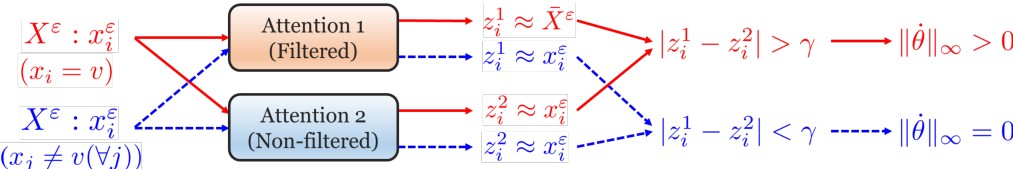

**Figure 3.** The adversarial server exploits self-attention mechanism to conduct inference attack of victim's protected local training data $D'$ in FL: If $x_i$ in the data equals to the target pattern $v$, the input to the filtered attention heads is the pertubed version $x_i^\varepsilon$ and the output $z_i^1$ of the filtered head is close to the protected pattern's average $\bar{X}^\varepsilon$ instead of $x_i^\varepsilon$. This creates non-zero gradients on weights computing on the difference of attention heads' outputs. The attack fails when the added noise is large and the embedding of protected data overlaps at the center of the embedding.

**Theorem 3.** *Given a $\Delta^\varepsilon$-separated data $D^{\mathcal{M}_\varepsilon}$ (the LDP-protected version of the data $D$) with i.i.d patterns of the security game $\mathsf{Exp}_{\mathrm{LDP}}^{\mathrm{AMI}}$, for any $\beta > 0$ large enough such that*

$$\Delta^\varepsilon \geq \frac{2}{\beta N_X} + \frac{1}{\beta}\log(2(N_X - 1)N_X \beta M^{\varepsilon 2}), \quad (5)$$

*there exists an AMI adversary, $\mathcal{A}_{\mathrm{Attn}}^{\mathcal{D}}$, that exploits the self-attention layer with a time complexity of $\mathcal{O}(d_X^3)$ such that $\mathbf{Adv}_{\mathrm{LDP}}^{\mathrm{AMI}}(\mathcal{A}_{\mathrm{Attn}}^{\mathcal{D}})$ is lower bounded by:*

$$\mathbf{Adv}_{\mathrm{LDP}}^{\mathrm{AMI}}(\mathcal{A}_{\mathrm{Attn}}^{\mathcal{D}}) \geq P_{\mathrm{proj}}^{\mathcal{D}^{\mathcal{M}_\varepsilon}}\left(\frac{1}{\beta N_X M^\varepsilon}\right)$$

$$+ P_{\mathrm{proj}}^{\mathcal{D}^{\mathcal{M}_\varepsilon}}\left(\frac{1}{\beta N_X M^\varepsilon}\right)^{2nN_X}$$

$$- P_{\mathrm{box}}^{\mathcal{D}^{\mathcal{M}_\varepsilon}}\left(3\bar{\Delta}^\varepsilon + \beta(m_{max}^\varepsilon)^2 R^\varepsilon\right) - 1 \quad (6)$$

*where $\bar{\Delta}^\varepsilon := 2M^\varepsilon(N_X - 1)\exp\left(2/N_X - \beta\Delta^\varepsilon\right)$ and $\mathcal{D}^{\mathcal{M}_\varepsilon}$ is the distribution of the protected data $D^{\mathcal{M}_\varepsilon}$ induced by the original data distribution $\mathcal{D}$ and the LDP-mechanism $\mathcal{M}_\varepsilon$. $m_x^\varepsilon = \frac{1}{N_X}\sum_{i=1}^{N_X} x_i^\varepsilon$ is the arithmetic mean of all LDP-protected patterns and $m_{max}^\varepsilon = \max_{1 \leq i \leq N_X} \|x_i - m_x^\varepsilon\|$. Here, $P_{\mathrm{proj}}^{\mathcal{D}^{\mathcal{M}_\varepsilon}}(\delta)$ is the probability that the projected component between two independent patterns drawn from $\mathcal{D}^{\mathcal{M}_\varepsilon}$ is smaller than $\delta$ and $P_{\mathrm{box}}^{\mathcal{D}^{\mathcal{M}_\varepsilon}}(\delta)$ is the probability that a random pattern drawn from $\mathcal{D}^{\mathcal{M}_\varepsilon}$ is in the cube of size $2\delta$ centering at the arithmetic mean of the patterns in $\mathcal{D}^{\mathcal{M}_\varepsilon}$. (Proof in Appx. D.4)*

The key step in proving Theorem 3 is to demonstrate that the configuration of the self-attention model, specified in Appendix C.2, behaves as outlined in Fig. 3. Given any input pattern $x_i \in X$, the attention heads process the LDP-protected version of the pattern, $x_i + r_i$. Let $v$ denote the target pattern. If $v$ is not present in the victim's training dataset, i.e., $x_j \neq v$ for all $1 \leq j \leq N_X$, then, using Lemma 1 (Appendix D.2), which builds on the *Exponentially Small Retrieval Error* Theorem for the attention layer (Ramsauer et al., 2021), we show that $z_1^h \approx x_i + r_i$ and $z_2^h \approx x_i + r_i$, both with a probability lower-bounded by:

$$P_{\mathrm{proj}}^{\mathcal{D}^{\mathcal{M}_\varepsilon}}\left(\frac{1}{\beta N_X M^\varepsilon}\right).$$

This probability governs the false-positive error of the attack.

If there exists $x_i = v$ (i.e., $v$ is in the training dataset), the weights of attention head 1 filter $v$ and output:

$$z_1^h = X^\varepsilon \mathrm{softmax}\left(\beta X^{\varepsilon\top}(r_i - \bar{r}_i^v)\right),$$

where $\bar{r}_i^v$ is the projection of $r_i$ onto $v$. If $R^\varepsilon$ is small enough, $z_1^h \approx \bar{X}^\varepsilon$. By computing the difference between the two heads, $|z_i^1 - z_i^2|$, the adversary can infer the presence of $v$ in $X$. For a hyperparameter $\gamma$, if $|z_i^1 - z_i^2| > \gamma$, then $v \in X$. Conversely, if $|z_i^1 - z_i^2| < \gamma$, then $v \notin X$. We select $\gamma = 2\bar{\Delta}^\varepsilon$, a choice justified in Appendix D.4.

False-negative errors occur when the embedding of protected data overlaps at the center of the embedding space, causing $|z_i^1 - z_i^2| < \gamma$ even when $v \in X$. By using the mean value theorem and bounding the Jacobian of the attention's forwarding function, this error is upper bounded by:

$$P_{\mathrm{box}}^{\mathcal{D}^{\mathcal{M}_\varepsilon}}\left(3\bar{\Delta}^\varepsilon + \beta(m_{\mathrm{max}}^\varepsilon)^2 R^\varepsilon\right).$$

As noise increases, the cube of size $6\bar{\Delta}^\varepsilon + 2\beta(m_{\mathrm{max}}^\varepsilon)^2 R^\varepsilon$ covers more patterns and causes $P_{\mathrm{box}}^{\mathcal{D}^{\mathcal{M}_\varepsilon}}$ to increase. At high noise levels, the attention outputs for all patterns are more likely to cluster near the center of the embedding space, as illustrated in Fig. 4. This leads to higher false-negative rates for the attack. However, models trained on such noisy data generally exhibit poor performance since patterns whose embeddings overlap in these central regions become indistinguishable. This results in $P_{\mathrm{box}}^{\mathcal{D}^{\mathcal{M}_\varepsilon}} \approx 1$ for large enough $R^\varepsilon$, causing the advantage to drop sharply regardless of dimensionality as the cube fully encloses the patterns. Our simulations of Eq. (6) for one-hot and spherical data are shown in Fig. 5 and Fig. 6, respectively.

**Remark 3.** $\Delta^\varepsilon$ **vs. the advantage's lower bound (6).** A larger $\Delta^\varepsilon$ allows a smaller $\beta$ to satisfy (5) and makes $P_{\mathrm{proj}}^{\mathcal{D}^{\mathcal{M}_\varepsilon}}\left(\frac{1}{\beta N_X M^\varepsilon}\right)$ larger and $P_{\mathrm{box}}^{\mathcal{D}^{\mathcal{M}_\varepsilon}}\left(3\bar{\Delta}^\varepsilon + \beta(m_{\mathrm{max}}^\varepsilon)^2 R^\varepsilon\right)$ smaller.

**Remark 4. The most vulnerable embedding.** The lower bound in (6) would be optimized with one-hot data.

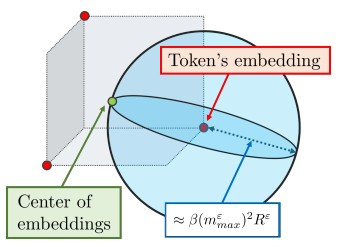

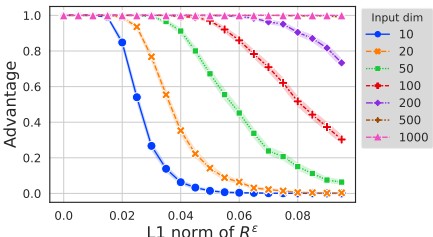

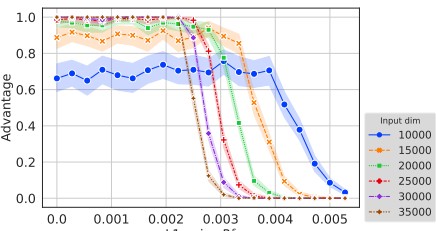

*Figure 4.* For large $R^\varepsilon$ s.t. $P_{\text{box}} \approx 1$, the embedding of protected data overlaps at the center of the embedding and impairs data utility.

*Figure 5.* $\mathbf{Adv}_{\text{LDP}}^{\text{AMI}}(\mathcal{A}_{\text{Attn}}^{\mathcal{D}})$ on one-hot data using Monte-Carlo simulation

*Figure 6.* $\mathbf{Adv}_{\text{LDP}}^{\text{AMI}}(\mathcal{A}_{\text{Attn}}^{\mathcal{D}})$ on spherical data using Monte-Carlo simulation

Since it has no alignment among patterns, $\Delta^\varepsilon$ achieves its maximum which is the pattern's norm. Furthermore, $P_{\text{proj}}^{\mathcal{D}^{\mathcal{M}_\varepsilon}}\left(\frac{1}{\beta N_X M^\varepsilon}\right)$ is 1 because all patterns are orthogonal to each other. Finally, since there is no pattern at the center of one-hot data, we can select a very large $\beta$ so that $P_{\text{box}}^{\mathcal{D}^{\mathcal{M}_\varepsilon}}\left(3\bar{\Delta}^\varepsilon + \beta(m_{\max}^\varepsilon)^2 R^\varepsilon\right) = 0$, given $R^\varepsilon$ is small enough.

**Remark 5. Asymptotic behavior of the advantage (6).** For high dimensional data, i.e., $d_X \to \infty$, two random points are surely almost orthogonal ($P_{\text{proj}}^{\mathcal{D}^{\mathcal{M}_\varepsilon}}\left(\frac{1}{\beta N_X M^\varepsilon}\right) \to 1$), and a random point is almost always at the boundary (Blum et al., 2020). Therefore, $P_{\text{box}}^{\mathcal{D}^{\mathcal{M}_\varepsilon}}\left(3\bar{\Delta}^\varepsilon + \beta(m_{\max}^\varepsilon)^2 R^\varepsilon\right) \to 0$ for small enough $R^\varepsilon$ and $\beta$. This phenomenon can be seen in one-hot data (see Fig. 5). For spherical data, the amount of noise needed for $P_{\text{box}}^{\mathcal{D}^{\mathcal{M}_\varepsilon}}\left(3\bar{\Delta}^\varepsilon + \beta(m_{\max}^\varepsilon)^2 R^\varepsilon\right) \to 1$ is smaller, and the cut-off noise's norm gradually decreases as the input dimension increases.

**Remark 6. Impact of $\beta$.** Increasing the hyper-parameter $\beta$ in $\mathcal{A}_{\text{Attn}}^{\mathcal{D}}$ (Algo. 3, Appx. C.2) would raise the memorization of the attention layer (Ramsauer et al., 2021). (Vu et al., 2024) showed experimentally that increasing $\beta$ leads to better adversarial success rates against unprotected data. For LDP-protected data, this is not the case, as increasing $\beta$ also increases $P_{\text{box}}^{\mathcal{D}^{\mathcal{M}_\varepsilon}}\left(3\bar{\Delta}^\varepsilon + \beta(m_{\max}^\varepsilon)^2 R^\varepsilon\right)$, reducing the lower bound of $\mathbf{Adv}_{\text{LDP}}^{\text{AMI}}(\mathcal{A}_{\text{Attn}}^{\mathcal{D}})$. We provide experiments on the impact of $\beta$ on $\mathbf{Adv}_{\text{LDP}}^{\text{AMI}}(\mathcal{A}_{\text{Attn}}^{\mathcal{D}})$ in Sect. 5.

## 5. Experiments

This section demonstrates the practical risks of leaking private data in FL. In particular, we implement the FC-based $\mathcal{A}_{\text{FC}}^{\mathcal{D}}$ and attention-based $\mathcal{A}_{\text{Attn}}^{\mathcal{D}}$ adversaries, and evaluate their success rates in synthetic and real-world datasets. Implementation details are given in Appendix. G

**Datasets and embedding.** Our experiments use two synthetic and three real-world datasets. The synthetic datasets include one-hot encoded data and spherical data (points

on the unit sphere). The real-world datasets, including CIFAR10, CIFAR100 (Krizhevsky et al., 2009), and ImageNet (Krizhevsky et al., 2012), are processed using pre-trained embedding modules to obtain data $D$ for our threat models. We also use the ImageNet dataset, a large-scale benchmark consisting of labeled images across 1,000 categories (Deng et al., 2009). For ResNet, we extract feature embeddings with Img2Vec (Safka, 2021), while for ViTs, we use pretrained foundation models provided by the authors on HuggingFace (Dosovitskiy et al., 2021). We refer readers to Appx. G.1 for more details.

**LDP mechanisms.** We use BitRand (Jiang et al., 2022), GRR (Warner, 1965), RAPPOR (Erlingsson et al., 2014), dBitFlipPM (Ding et al., 2017) as LDP mechanisms for real-world datasets. Details about these algorithms are in Appx. G.4. We also provide some results on OME (Lyu et al., 2020) in Appx. H.1.

**Results on synthetic datasets.** Fig. 5 and Fig. 6 shows the impact of the L1 norm of $R^\varepsilon$ on the advantage of Attention-based AMI adversary on one-hot and spherical data, respectively. For one-hot data, as we increase $d_X$, random noise is almost always at the boundary and there is no pattern at the center of one-hot data. This reduces the likelihood that the protected data's embedding overlaps with the center of the embeddings, increasing the lower bound of the adversary's advantage in (6). On the other hand, for spherical data, the advantage drops sharply when $R^\varepsilon$ increases, regardless of the dimension of the data.

**Results of FC-based AMI adversary.** Figures 7 and 8 show the success rates of FC-based AMI attacks against 4 different LDP algorithms on CIFAR10 and CIFAR100. The theoretical lower and upper bound on the adversary's attack success rate can be derived directly from the theoretical lower and upper bound of $\mathbf{Adv}_{\text{LDP}}^{\text{AMI}}(\mathcal{A}_{\text{FC}}^{\mathcal{D}})$ using Eq. 1. Across these LDP mechanisms, the amount of LDP noise needed to protect against AMI attack significantly reduces the model's utility. For example, for BitRand-protected CIFAR10, to make the inference rate lower than 80% (yellow

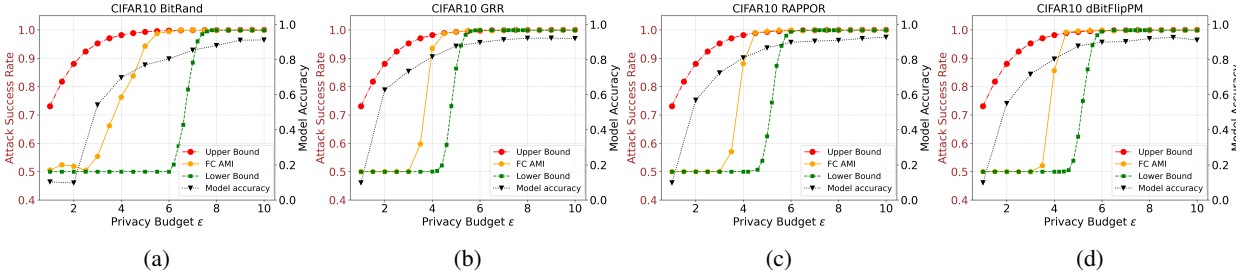

Figure 7. Theoretical upper/lower bound and empirical results on the attack success rates of FC-based AMI adversaries against CIFAR10 dataset protected by BitRand (a), GRR (b), RAPPOR (c) and dBitFlipPM (d).

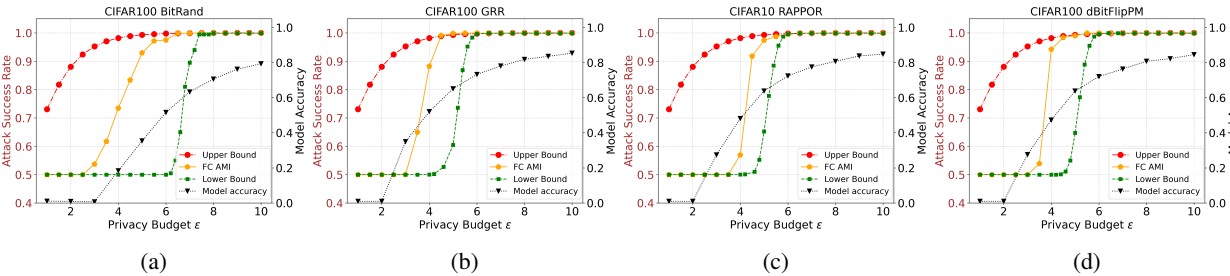

Figure 8. Theoretical upper/lower bound and empirical results on the attack success rates of FC-based AMI adversaries against CIFAR100 dataset protected by BitRand (a), GRR (b), RAPPOR (c) and dBitFlipPM (d).

line), the model has to suffer at least 20% accuracy loss. The theoretical lower bound of the attack's success rate corroborates the empirical success rate of $\approx 100\%$ when $\epsilon = 8$. For both real-world datasets, $\mathcal{A}_{\mathsf{FC}}^{\mathcal{D}}$ achieves near 100% success rate when $\epsilon$ approaches 6. Among the evaluated mechanisms, GRR, RAPPOR, and dBitFlipPM preserve higher model accuracy at lower $\varepsilon$ values but expose the model to greater privacy risks, as indicated by both the empirical and theoretical attack success rates approaching 100% at $\varepsilon = 5$ and 6, respectively.

**Results of Attention-based AMI adversary.** Experiments on CIFAR10 and ImageNet (1000 classes) are conducted on ViT-B-32-224 and ViT-B-32-384, respectively (Fig. 9). For LDP-protected data, the inference success rate of $\mathcal{A}_{\mathsf{Attn}}^{\mathcal{D}}$ approaches 100% when $\epsilon = 3$ or higher. At this privacy budget, model performance suffers significantly. Fig. 9 also illustrates the impact of batch size on the attack success rates, showing that the proposed $\mathcal{A}_{\mathsf{Attn}}^{\mathcal{D}}$ performs consistently across different batch sizes.

**Impact of $\beta$ on $\mathbf{Adv}_{\mathbf{LDP}}^{\mathbf{AMI}}(\mathcal{A}_{\mathsf{Attn}}^{\mathcal{D}})$.** As discussed in remark 6, increasing $\beta$ also increases $P_{\mathrm{box}}^{\mathcal{D}^{\mathcal{M}_\varepsilon}}\left(3\bar{\Delta}^\varepsilon + \beta(m_{\max}^\varepsilon)^2 R^\varepsilon\right)$, and in turn makes the lower bound of Eq. (6) smaller. We illustrate this behavior in Fig.10. The more we increase $\beta$, the less likely the adversary succeed. Note that we still need to choose a $\beta$ large enough for Eq. 5 to hold. We further discuss the choice of hyperparameters in Appendix G.3. Given the assumption that the server has knowledge of the

client's data distribution (as outlined in the AMI threat models in Section 3.1), the server can simulate the client's data to compute a minimally sufficient value for $\beta$. When doing experiments, we found that setting $\beta$ to a reasonably small value (e.g., 0.01) yielded consistently good results across realistic $\varepsilon$ values and datasets/LDP mechanisms. As illustrated in Figure 10, for LDP-protected data, $\beta = 0.01$ generally achieves better success rates under small $\varepsilon$.

**Empirical results on NLP datasets.** In section 4.2, the distortion imposed by LDP is modeled by a noise $r_i$ added to each pattern: $X^\varepsilon = \mathcal{M}^\varepsilon(X) = \{x_i + r_i\}_{i=1}^{N_X} = \{x_i^\varepsilon\}_{i=1}^{N_X}$. In our analysis, we assume $x_i$ and $r_i$ to be continuous, and the impact of LDP noise can be visualized in Fig. 4. For NLP data, both the data and the noise should be modeled as discrete, hence our theoretical analysis might not directly apply to the NLP scenario. The key challenge is that in NLP, tokens are typically represented as discrete embeddings, and adding continuous noise is not meaningful in this context. Therefore, a separate theoretical framework would be required to account for the discrete nature of NLP data.

However, it is important to note that the attack still experimentally works against both vision and NLP data. To demonstrate this, we conducted comprehensive experiments across 4 NLP datasets (IMDB (Maas et al., 2011), Yelp (Zhang et al., 2015), Twitter (Saravia et al., 2018), Finance (Casanueva et al., 2020)), 4 models (BERT (Devlin et al., 2019), RoBERTa (Liu et al., 2019), GPT-1 (Radford et al.,

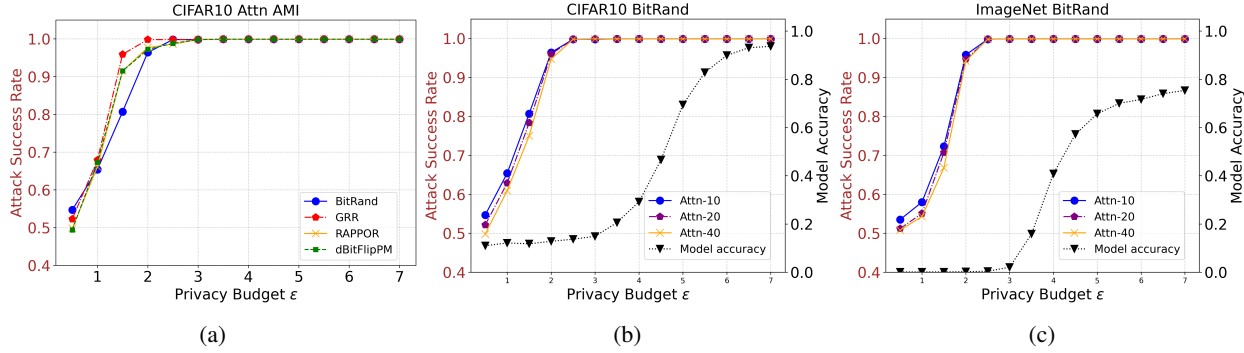

*Figure 9.* Comparison of success rates of Attention-based AMI adversaries against CIFAR10 protected by BitRand, GRR, RAPPOR, and dBitFlipPM (a) as well as the privacy-utility trade-off and the impact of batch size on attack success on BitRand-protected data (b,c). Here, Attn-10 means the attack is conducted with batch size 10, and so on. Batch size is 10 if not explicitly mentioned.

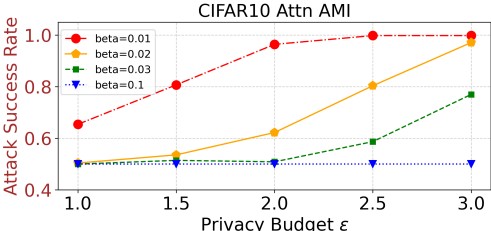

*Figure 10.* Impact of $\beta$ on $\mathbf{Adv}_{\mathrm{LDP}}^{\mathrm{AMI}}(\mathcal{A}_{\mathrm{Attn}}^{\mathcal{D}})$

2018), DistilBERT (Sanh et al., 2019)), and 3 LDP algorithms (GRR, RAPPOR, dBitFlipPM). The results given in Appx. H.2 indicate that privacy risks persist even for LLMs, depending on the privacy budget. To explore more in depth the impact of different LDP mechanisms on the attack success rates, we also conduct an ROC analysis of the attack success rates (on IMDB dataset) in Appx. H.3.

## 6. Conclusion

This work studies the formal threat models for AMI attacks with dishonest FL servers, effectively and rigorously providing the theoretical bound on the vulnerabilities of FL under LDP protection. We also provide experimental evidence for the high success rates of active inference attacks under certain LDP mechanisms. The results imply that LDP-protected data might be vulnerable to inference attacks in FL with dishonest servers and clients should carefully consider the tradeoff between privacy and utility.

## Acknowledgments

This work was supported in part by the National Science Foundation under grants III-2416606 and SaCT-1935923.

This work was authored in part by the National Renewable Energy Laboratory (NREL) for the U.S. Department of Energy (DOE) under Contract No. DE-AC36-08GO28308. Funding provided by the Laboratory Directed Research and Development (LDRD) Program at NREL. The views expressed in the article do not necessarily represent the views of the DOE or the U.S. Government. The U.S. Government retains and the publisher, by accepting the article for publication, acknowledges that the U.S. Government retains a nonexclusive, paid-up, irrevocable, worldwide license to publish or reproduce the published form of this work, or allow others to do so, for U.S. Government purposes.

## Impact Statement

Our findings demonstrate that LDP-protected data can still be compromised by dishonest FL servers, particularly under practical scenarios. Additionally, the study highlights the significant trade-offs between privacy and model utility, showing that the noise levels necessary to mitigate these attacks often lead to substantial degradation in model performance.

Our work has immediate implications for the design and deployment of FL systems, emphasizing the need for stronger privacy safeguards and more robust defenses against AMI attacks. Researchers and practitioners are encouraged to explore enhanced privacy-preserving techniques that balance security with model effectiveness. Furthermore, policymakers can leverage these insights to establish clearer privacy guidelines for the implementation of FL in sensitive applications, such as healthcare and finance.

By providing a theoretical framework alongside empirical evidence, our paper serves as a critical resource for understanding and addressing privacy threats in decentralized machine learning environments. The results pave the way for further advancements in secure and privacy-preserving AI technologies.

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

## A. Appendix

This is the appendix of our paper *Theoretically Unmasking Inference Attacks Against LDP-Protected Clients in Federated Vision Models*. Its main content and outline are as follows:

  – Appendix D.1: the proof of Theorem 1.
  – Appendix D.2 discusses attention layer's memorization capabilities through *Exponentially Small Retrieval Error Theorem* (Ramsauer et al., 2021).
  – Appendix D.3: the impact of LDP mechanisms on data separation.
  – Appendix D.4: the proof of Theorem 3.

- Appendix I discusses alternative privacy-preserving techniques such as SMPC and Homomorphic Encryption.

## B. Active Inference Threat Models as Security Games

This appendix provides the descriptions of the security games examined in our work. All games are conducted between a challenger/client, and an adversary/server in FL. The adversary is denoted by $\mathcal{A}^{\mathcal{D}}$, in which the superscript $\mathcal{D}$ indicates that the server knows the data distribution of the client's private data. At the beginning of the games, a random bit $b$ is generated and it is used to decide whether the challenger's private data has a specific sample. The goal of the AMI adversary is to guess the bit $b$, which is equivalent to inferring information on the challenger's data.

As pointed out briefly in Sect. 3.1, the adversarial server $\mathcal{A}^{\mathcal{D}}$ in all security games consists of three components $\mathcal{A}^{\mathcal{D}}_{\text{INIT}}$, $\mathcal{A}^{\mathcal{D}}_{\text{ATTACK}}$ and $\mathcal{A}^{\mathcal{D}}_{\text{GUESS}}$. An illustration of their dynamics is provided in Fig. 1. To specify an adversary, for each security game, we need to describe how it determines the model $\Phi$ for FL in $\mathcal{A}^{\mathcal{D}}_{\text{INIT}}$, how it crafts the model's parameters $\theta$ in $\mathcal{A}^{\mathcal{D}}_{\text{ATTACK}}$ and how it guesses the bit $b$ in $\mathcal{A}^{\mathcal{D}}_{\text{GUESS}}$. The security games considered in this work are described below.

**AMI on unprotected data** $\text{Exp}^{\text{AMI}}_{\text{NONE}}(\mathcal{A}^{\mathcal{D}})$: This security game is about the base AMI threat model, which is first formulated in (Nguyen et al., 2023) to study the threat of AMI in FL. While we do not directly study this security game, it serves as the foundation for $\text{Exp}^{\text{AMI}}_{\text{LDP}}(\mathcal{A}^{\mathcal{D}})$. The subscript NONE indicates there is no defense mechanism applied on the data. The goal of the adversary is to decide if a target sample $T$ is included in the training data $D$. Fig. 11 provides the pseudo-code of this security game.

$\underline{\mathsf{Exp}^{\mathrm{AMI}}_{\mathrm{NONE}}(\mathcal{A}^{\mathcal{D}})}$:

# Simulating the dataset $D$ of the client
$D \leftarrow \emptyset$
**while** $|D| < n$ **do**
    $X \xleftarrow{\mathcal{D}} \mathcal{X}$ # Sampling $X$ from the input distribution $\mathcal{D}$
    **if** $X \notin D$ **then**
        $D \leftarrow D \cup \{X\}$
# The random bit game
$b \xleftarrow{\$} \{0,1\}$
$X \leftarrow \textsc{None}$
**if** $b = 1$ **then**
    $T \xleftarrow{\$} D$ # Uniformly sampling $T$ from $n$ samples in $D$
**else**
    **while** $T == \textsc{None}$ or $T \in D$ **do**
        $T \xleftarrow{\mathcal{D}} \mathcal{X}$ # Sampling $T$ from the input distribution
# The attack
$\Phi \leftarrow \mathcal{A}^{\mathcal{D}}_{\mathrm{INIT}}$ # The adversarial server decides a model $\Phi$
$\theta \leftarrow \mathcal{A}^{\mathcal{D}}_{\mathrm{ATTACK}}(T)$ # The server computes the parameters $\theta$ based on the target input $T$
$\dot{\theta} \leftarrow \nabla_\theta \mathcal{L}_\Phi(D)$ # The client computes the gradients of the loss w.r.t the parameters based on its data $D$
$b' \leftarrow \mathcal{A}^{\mathcal{D}}_{\mathrm{GUESS}}(T, \dot{\theta})$ # The server receives $\dot{\theta}$ and guesses a bit $b'$
**Ret** $[b' = b]$ # The game returns 1 if $b' = b$ (the adversarial server wins), 0 otherwise

*Figure 11.* The AMI Threat Model as a Security Game.

$\underline{\mathsf{Exp}^{\mathrm{AMI}}_{\mathrm{LDP}}(\mathcal{A}^{\mathcal{D}}, \varepsilon)}$:

# Simulating the dataset $D$ of the client
As in the AMI threat model in Fig. 11
# The random bit game
As in the AMI threat model in Fig. 11
# The attack
$\Phi \leftarrow \mathcal{A}^{\mathcal{D}}_{\mathrm{INIT}}$
$\theta \leftarrow \mathcal{A}^{\mathcal{D}}_{\mathrm{ATTACK}}(T)$
$D' \leftarrow \mathcal{M}^\varepsilon(D) = \{\mathcal{M}^\varepsilon(X)\}_{X \in D}$ # Applying LDP mechanism $\mathcal{M}$ with $\varepsilon$ parameter on the data $D$
$\dot{\theta} \leftarrow \nabla_\theta \mathcal{L}_\Phi(D')$ # Gradients are computed on the protected data
$b' \leftarrow \mathcal{A}^{\mathcal{D}}_{\mathrm{GUESS}}(T, \dot{\theta})$
**Ret** $[b' = b]$

*Figure 12.* The AMI threat model under LDP mechanism as a security game.

**AMI on LDP-protected data** $\mathsf{Exp}^{\mathrm{AMI}}_{\mathrm{LDP}}(\mathcal{A}^{\mathcal{D}})$: This security game describes the AMI threat model when the data is protected by LDP mechanisms. The work (Nguyen et al., 2023) extends $\mathsf{Exp}^{\mathrm{AMI}}_{\mathrm{NONE}}(\mathcal{A}^{\mathcal{D}})$ to obtain the formulation of $\mathsf{Exp}^{\mathrm{AMI}}_{\mathrm{LDP}}(\mathcal{A}^{\mathcal{D}})$. In this game, the client independently perturbs its training data sample $D$ using an LDP-preserving mechanism $\mathcal{M}$ to obtain a randomized local training set $D' = \mathcal{M}^\varepsilon(D) = \{\mathcal{M}^\varepsilon(X)\}_{X \in D}$. This randomized data $D'$ is then used for the local training instead of $D$. As a result, the gradients that the FL server receives are computed on the protected data $D'$ instead of $D$. Fig. 12 is the pseudo-code of this security game.

# C. FC-based and Attention-based AMI Attacks

This appendix reports the details of FC-based and Attention-based AMI Attacks. In Appx. C.1, we provide the descriptions of the FC-based AMI adversary proposed by (Vu et al., 2024) used in our analysis. Appx. C.2 presents the details of the Attention-based AMI adversary (Vu et al., 2024).

## C.1. FC-Based Adversary for AMI in FL

We now describe the AMI FC-based adversary $\mathcal{A}_{\mathsf{FC}}^{\mathcal{D}}$ proposed by (Vu et al., 2024) and mentioned in Sect. 3.2. The adversary consists of 3 components $\mathcal{A}_{\mathsf{FC-INIT}}^{\mathcal{D}}$, $\mathcal{A}_{\mathsf{FC-ATTACK}}^{\mathcal{D}}$ and $\mathcal{A}_{\mathsf{FC-GUESS}}^{\mathcal{D}}$.

---

**Algorithm 1** $\mathcal{A}_{\mathsf{FC-ATTACK}}^{\mathcal{D}}(T)$ exploiting fully-connected layer in AMI

---

**Hyper-parameters:** $\tau^{\mathcal{D}} \in \mathbb{R}^+$

1 # Configuring $W_1 \in \mathbb{R}^{2d_X \times d_X}$ and $b_1 \in \mathbb{R}^{2d_X}$ of the first FC

$$W_1 \leftarrow \begin{bmatrix} I_{d_X} \\ -I_{d_X} \end{bmatrix}, \quad b_1 \leftarrow \begin{bmatrix} -T \\ T \end{bmatrix}$$

2 # Configuring the first row of $W_2 \in \mathbb{R}^{d \times 2d_X}$ and the first entry of $b_2 \in \mathbb{R}^d$ of the second FC

$$W_2[1,:] \leftarrow -1_{2d_X}^\top, \quad b_2[1] \leftarrow \tau^{\mathcal{D}}$$

3 **Ret** all weights and biases

---

---

**Algorithm 2** $\mathcal{A}_{\mathsf{FC-GUESS}}^{\mathcal{D}}(T, \dot{\theta})$ exploiting fully-connected layer in AMI

---

# If the gradient of $b_2[1]$ is non-zero, returns 1

**if** $|\dot{\theta}(b_2[1])| > 0$ **then**
  |   **Ret** 1
**end**
**Ret** 0

---

**AMI initialization** $\mathcal{A}_{\mathsf{FC-INIT}}^{\mathcal{D}}$: The adversary's model employs fully connected (FC) layers for its first two layers. Given an input $X \in \mathbb{R}^{d_X}$, the attacker computes $\mathrm{ReLU}(W_l X + b_l) = \max(0, W_l X + b_l)$, where $W_l$ and $b_l$ denote the weights and biases of layer $l$, respectively. The dimensions of $W_1$ and $b_1$ are set to $2d_X \times d_X$ and $2d_X$, respectively. For the second layer, the attack only analyzes a single output neuron, thus, requiring $W_2$ to have only $2d_X$ columns. We denote the parameters associated with this neuron as $W_2[1,:]$ and $b_2[1]$. Models with additional parameters can still works, as surplus parameters can simply be disregarded.

**AMI attack** $\mathcal{A}_{\mathsf{FC-ATTACK}}^{\mathcal{D}}$: The weights and biases of the first two FC layers are set as:

$$W_1 \leftarrow \begin{bmatrix} I_{d_X} \\ -I_{d_X} \end{bmatrix}, \quad b_1 \leftarrow \begin{bmatrix} -T \\ T \end{bmatrix}, \quad W_2[1,:] \leftarrow -1_{d_X}^\top, \quad b_2[1] \leftarrow \tau^{\mathcal{D}} \tag{7}$$

where $I_{d_X}$ is the identity matrix and $1_{d_X}$ is the all-ones vector of size $d_X$. The hyperparameter $\tau^{\mathcal{D}}$ controls the total allowable distance between an input $X$ and the target $T$, which can be determined from the distribution statistics. The pseudo-code of the attack is presented in Algo. 1.

**AMI guess** $\mathcal{A}_{\mathsf{FC-GUESS}}^{\mathcal{D}}$: In the guessing phase, the AMI server returns 1 if the gradient of $b_2[1]$ is non-zero, and returns 0 otherwise. The pseudo-code of this step is shown in Algo. 2.

## C.2. Attention-Based Adversary for AMI in FL

We now describe the AMI attention-based adversary $\mathcal{A}_{\mathsf{Attn}}^{\mathcal{D}}$ introduced in (Vu et al., 2024) and discussed in Sect. 3.3. The 3 components of it are $\mathcal{A}_{\mathsf{Attn-INIT}}^{\mathcal{D}}$, $\mathcal{A}_{\mathsf{Attn-ATTACK}}^{\mathcal{D}}$ and $\mathcal{A}_{\mathsf{Attn-GUESS}}^{\mathcal{D}}$, detailed as follows:

**AMI initialization** $\mathcal{A}_{\mathsf{Attn-INIT}}^{\mathcal{D}}$: The model initiated by the dishonest server has self-attention as its first layer. We set the number of attention heads $H$ to 4. The attention dimension is $d_{\mathsf{attn}} = d_X - 1$, where $d_X$ is the one-hot encoding dimension. The hidden and output dimensions are $d_{\mathsf{hid}} = d_X$ and $d_Y = 2d_X$, respectively. Any configurations with a higher number of parameters can adopt the proposed attack because the extra parameters can simply be ignored.

---

**Algorithm 3** $\mathcal{A}^{\mathcal{D}}_{\text{Attn}-\text{ATTACK}}(v)$ using self-attention in AMI

---

**Hyper-parameters:** $\beta, \gamma \in \mathbb{R}^+$

4  Randomly initialize $W^h_Q, W^h_K, W^h_V, W_O$ and $b_O$ for all heads $h$

5  Randomly initialize a matrix $W \in \mathbb{R}^{d_X \times d_X}$

6  $W[:, 1] \leftarrow v$ # Set the first column of $W$ to $v$

7  $Q, R \leftarrow \text{QR}(W)$ # QR-factorization $W$

8  $W^1_Q \leftarrow Q[2 : d_X]^\top$ # Set $W^1_Q$ to the last $d_X - 1$ rows of $Q^\top$

9  $W^1_K \leftarrow \beta W^{1\dagger\top}_Q$ # Set head 1 to memorization mode

10 $W^2_K \leftarrow \beta W^{2\dagger\top}_Q$ # Set head 2 to memorization mode

11 $W^3_Q \leftarrow W^1_Q, W^3_K \leftarrow W^1_K$ # Copy Head 1 to Head 3

12 $W^4_Q \leftarrow W^2_Q, W^4_K \leftarrow W^2_K$ # Copy Head 2 to Head 4

13 $W^1_V \leftarrow I_{d_X}, W^2_V \leftarrow I_{d_X}, W^3_V \leftarrow I_{d_X}, W^4_V \leftarrow I_{d_X}$ # Setup for detection

14 $W_O \leftarrow \begin{bmatrix} I_{d_X} & -I_{d_X} & 0_{d_X} & 0_{d_X} \\ 0_{d_X} & 0_{d_X} & -I_{d_X} & I_{d_X} \end{bmatrix}$ # Setup for detection

15 $b_{O i} = -\gamma, \forall i \in \{1, \cdots, d_Y\}$ # Setup biases

16 **Ret** all weights and biases

---

**Algorithm 4** $\mathcal{A}^{\mathcal{D}}_{\text{Attn}-\text{GUESS}}(v, \dot{\theta})$ using self-attention in AMI

---

# If the any gradients of $W^O$ is non-zero, returns 1

**if** $\|\dot{\theta}_1(W^O)\|_\infty > 0$ **then**
 | **Ret** 1
**end**
**Ret** 0

---

**AMI attack $\mathcal{A}^{\mathcal{D}}_{\text{Attn}-\text{ATTACK}}$:** The attack component $\mathcal{A}^{\mathcal{D}}_{\text{Attn}-\text{ATTACK}}$ determines the self-attention weights including $W^h_Q, W^h_K, W^h_V, W_O$ and $b_O$, where $h$ is the head's index. There are two hyper-parameters, $\beta$ and $\gamma \in \mathbb{R}^+$, in $\mathcal{A}^{\mathcal{D}}_{\text{Attn}-\text{ATTACK}}$. Intuitively, $\beta$ controls how much the attention heads memorize their input patterns $x^h_i$ and $\gamma$ adjusts a cut-off threshold deciding between $v \in D$ and $v \notin D$ (depicted in Fig. 3). Given a target pattern $v$ for detection, the weights of the first attention head is chosen such that:

$$W^{1\top}_K W^1_Q \approx \beta I_{d_X} \quad \text{and} \quad W^1_Q v \approx 0 \tag{8}$$

To enforce condition (8), $d_X - 1$ vectors orthogonal to $v \in \mathbb{R}^{d_X}$ are assigned to $W^1_Q$ using QR-factorization. Subsequently, $W^1_K$ is defined as the transpose of $\beta W^{1\dagger}_Q$, where $\dagger$ represents the pseudo-inverse. In contrast, the second head initializes $W^2_Q$ randomly and sets $W^2_K$ as its pseudo-inverse. As a result, the second condition of (8) does not hold for $W^2_Q$ and $W^2_K$. The remaining parameters of the two heads are configured such that the first $d_X$ rows of $Y$ compute $\max\{0, Z^1 - Z^2 - \gamma \mathbf{1}^\top\}$. The third and fourth heads are designed to produce the negation of the first and second heads, respectively, meaning the last $d_X$ rows of $Y$ compute $\max\{0, Z^2 - Z^1 - \gamma \mathbf{1}^\top\}$. For simplicity in analysis, $W^h_V$ and $W^O$ are constructed using identity and zero matrices. The pseudo-code of the attack is provided in Algo. 3.

**AMI guess $\mathcal{A}^{\mathcal{D}}_{\text{Attn}-\text{GUESS}}$:** In the guessing phase, the AMI server checks if any of the weights in $W_O$ have non-zero gradients. Algo. 4 shows the pseudo-code of this step.

**Attack strategy.** We analyze the attack strategy of $\mathcal{A}^{\mathcal{D}}_{\text{Attn}}$ against unprotected data. The attention-based attacks exploit the memorization capability of the attention layer (Ramsauer et al., 2021): the attacks determine the layer's weights so that if a pattern $\xi$ similar to a stored pattern $x \in X$ feed to the layer, the returned signal will be similar to the stored pattern $x$. $X$ can be considered as the Key and the Value, while $\xi$ is the Query in the attention mechanism. The memorization is imposed by the condition $W^{1\top}_K W^1_Q \approx \beta I_{d_X}$ in (8). For an input $X$ that does not contain the target pattern $v$, we have $1/\beta X^\top W^{1\top}_K W^1_Q X \approx X^\top X$, which is the matrix of correlations of patterns in $X$. The output of the softmax, therefore, approximates $I_{N_X}$ since the diagonal entries of $X^\top X$ are significantly larger than the non-diagonal entries. The head's output $Z^1 \approx X$, i.e., $z^1_i \approx x_i$, as a result. Since the second head is the same as the first for $x \neq v$, we also have $Z^2 \approx x_i$. When $X$ contains the target pattern $v$, i.e., there exists an $x_i$ such that $x_i = v$, due to the second condition of Eq.(8), we have

$x_i^\top W_K^{1\top} W_Q^1 x_i \approx 0$. This makes the softmax's output uniform, which can be interpreted as the attention being distributed equally among all patterns. Consequentially, the attention's output of the first head is the pattern's average $\bar{X}$. Since the second head does not filter $v$, its output approximates $x_i$. By computing the difference between the two heads $|z_i^1 - z_i^2|$ and offset it by $\gamma$, the adversary can infer the presence of $v$ in $X$. While the attack strategy of $\mathcal{A}_{\text{Attn}}^{\mathcal{D}}$ against LDP-protected data follows the same principle, the main difference is the input of the attention heads is now the protected version $x_i^\varepsilon$ instead of $x_i$ like in the case of unprotected data. The introduced noise means that $(x_i^\varepsilon)^\top W_K^{1\top} W_Q^1 x_i^\varepsilon$ might not be $\approx 0$, making the softmax's output non-uniform, and the attention's output is no longer the pattern's average. We analyze in-depth the behavior of $\mathcal{A}_{\text{Attn}}^{\mathcal{D}}$ under LDP in Appendix D.4.

## D. Vulnerability of FL under LDP to AMI

This appendix provides the details of our theoretical results on AMI in FL under LDP. Appendix D.1 shows the proof of Theorem 1, which is about the vulnerability of LDP-protected data against FC-based AMI attack. Appendix D.2 analyzes the memorization capabilities of attention layers. An example demonstrating the impact of LDP mechanisms on the data's separation is provided in Appendix D.3. Finally, Appendix D.4 shows the proof of Theorem 3, which is about the vulnerability of LDP-protected data against attention-based AMI attack.

### D.1. Proof of Theorem 1 on the Vulnerability of LDP-Protected Data to AMI in FL

This appendix provides the proof of Theorem 1, which is restated below:

**Theorem.** *Given the security game* $\mathsf{Exp}_{\text{LDP}}^{\text{AMI}}$, *there exists an AMI adversary* $\mathcal{A}_{\text{FC}}^{\mathcal{D}}$ *whose time complexity is* $\mathcal{O}(d_X^2)$ *such that* $\mathbf{Adv}_{\text{LDP}}^{\text{AMI}}(\mathcal{A}_{\text{FC}}^{\mathcal{D}}) \geq 1 - \frac{n + |\mathcal{X}| - 1}{|\mathcal{X}| - 1} P_{\mathcal{M}^\varepsilon}$, *where* $n$ *is the size of the dataset* $D$, $|\mathcal{X}|$ *is the cardinality of the possible output values of the LDP-mechanism and* $P_{\mathcal{M}^\varepsilon}$ *is the probability that the LDP-mechanism makes the protected version of data point inside the neighborhood of another data point.*

*Proof.* Given $D = \{X_i\}_{i=1}^n$, where $X_i \in \mathcal{X}$, and $\mathcal{X} \subseteq \mathbb{R}^{d_X}$, the LDP-protected version of the input $X$ is $\mathcal{M}^\varepsilon(X)$. For the model specified by $\mathcal{A}_{\text{FC}}^{\mathcal{D}}$ as discussed in Subsection 3.2 and described in Appendix C.1, its first layer computes:

$$\text{ReLU}\left(\begin{bmatrix} I_{d_X} \\ -I_{d_X} \end{bmatrix} \mathcal{M}^\varepsilon(X) + \begin{bmatrix} -T \\ T \end{bmatrix}\right) = \text{ReLU}\left(\begin{bmatrix} \mathcal{M}^\varepsilon(X) - T \\ T - \mathcal{M}^\varepsilon(X) \end{bmatrix}\right) \tag{9}$$

The first row of the second layer then computes:

$$z_0 := \text{ReLU}\left(-\sum_{i=1}^{d_X} \text{ReLU}\left((x_i^\varepsilon - t_i) + \text{ReLU}(t_i - x_i^\varepsilon)\right) + \tau^{\mathcal{D}}\right) = \max\left\{\tau^{\mathcal{D}} - \|\mathcal{M}^\varepsilon(X) - T\|_{L_1}, 0\right\} \tag{10}$$

This implies the gradient of $b_2[1] = \tau^{\mathcal{D}}$ is non-zero if and only if $\tau^{\mathcal{D}} > \|\mathcal{M}^\varepsilon(X) - T\|_{L_1}$. Thus, for a small enough $\tau^{\mathcal{D}}$, $T \in D$ is equivalent to a non-zero gradient. We set $\tau^{\mathcal{D}} = \Delta^{\mathcal{X}}$. Intuitively, the attack fails if either: (1) $T \notin D$, but $\exists X \in D$ such that $\mathcal{M}(X) \in B_1(T, \Delta^{\mathcal{X}})$ or (2) $T \in D$, but $\mathcal{M}(X) \notin B_1(T, \Delta^{\mathcal{X}})$ for all $X \in D$. This insight is illustrated in Fig. 2.

Given a data $D$ and a randomly sampled point $T \notin D$, the probability that $\mathcal{M}^\varepsilon(X)$ belongs to $B_1(T, \Delta^{\mathcal{X}})$ is upper bounded by:

$$\Pr\left[\mathcal{M}^\varepsilon(X) \in B_1(T, \Delta^{\mathcal{X}})\right] = \Pr\left[\mathcal{M}^\varepsilon(X) \in B_1(T, \Delta^{\mathcal{X}}) \text{ and } \mathcal{M}^\varepsilon(X) \notin B_1(X, \Delta^{\mathcal{X}})\right] \tag{11}$$

$$= \Pr\left[\mathcal{M}^\varepsilon(X) \in B_1(T, \Delta^{\mathcal{X}}) | \mathcal{M}^\varepsilon(X) \notin B_1(X, \Delta^{\mathcal{X}})\right] \Pr\left[\mathcal{M}^\varepsilon(X) \notin B_1(X, \Delta^{\mathcal{X}})\right] \tag{12}$$

$$\leq \frac{1}{|\mathcal{X}| - 1} \Pr\left[\mathcal{M}^\varepsilon(X) \notin B_1(X, \Delta^{\mathcal{X}})\right] = \frac{1}{|\mathcal{X}| - 1} P_{\mathcal{M}^\varepsilon} \tag{13}$$

(11) is from the fact that $\mathcal{M}^\varepsilon(X) \in B_1(T, \Delta^{\mathcal{X}})$ implies $\mathcal{M}^\varepsilon(X) \notin B_1(X, \Delta^{\mathcal{X}})$ as the balls are disjoint. (12) is from the conditional probability formula. For (13), given $\mathcal{M}^\varepsilon(X) \notin B_1(X, \Delta^{\mathcal{X}})$, since all the balls $B_1(X, \Delta^{\mathcal{X}})$ for $X \in D$ and $B_1(T, \Delta^{\mathcal{X}})$ are disjoint, $\mathcal{M}^\varepsilon(X)$ can either be in one of the other $|\mathcal{X}| - 1$ balls of radius $\Delta^{\mathcal{X}}$ or be outside of all those balls. As $T$ are chosen randomly, the probability that $\mathcal{M}^\varepsilon(X)$ is in one of the $|\mathcal{X}| - 1$ balls is bounded by $1/(|\mathcal{X}| - 1)$.

If $b = 0$, the probability that $z_0$ is activated is:

$$\Pr\left[z_0 > 0 | b = 0\right] = \Pr\left[\exists X \in D \text{ such that } \mathcal{M}(X) \in B_1(T, \Delta^{\mathcal{X}}) | b = 0\right] \tag{14}$$

$$\leq \sum_{X \in D} \frac{1}{|\mathcal{X}| - 1} P_{\mathcal{M}^\varepsilon} \leq \frac{n}{|\mathcal{X}| - 1} P_{\mathcal{M}^\varepsilon} \tag{15}$$

where (14) is from (10) and the inequalities in (15) are from the union bound and (13).

On the other hand, if $b = 1$, the probability that $z_0$ is not activated is bounded by:

$$\Pr\left[z_0 = 0 | b = 1\right] = \Pr\left[\mathcal{M}(X) \notin B_1(T, \Delta^{\mathcal{X}}) \text{ for all } X \in D | b = 1\right] \tag{16}$$

$$\leq \Pr\left[\mathcal{M}(T) \notin B_1(T, \Delta) | b = 1\right] = \Pr\left[\mathcal{M}(T) \notin B_1(T, \Delta)\right] = P_{\mathcal{M}^\varepsilon} \tag{17}$$

where inequality (17) uses the fact that, conditioned on $b = 1$, $T \in D$.

Thus, we have the advantage of $\mathcal{A}_{\mathsf{FC}}^{\mathcal{D}}$:

$$\mathbf{Adv}_{\mathrm{LDP}}^{\mathrm{AMI}}(\mathcal{A}_{\mathsf{FC}}^{\mathcal{D}}) = \Pr[b' = 1 | b = 1] + \Pr[b' = 0 | b = 0] - 1 \tag{18}$$

$$= (1 - \Pr[z_0 = 0 | b = 1]) + (1 - \Pr[z_0 > 0 | b = 0]) - 1 \tag{19}$$

$$\geq (1 - P_{\mathcal{M}^\varepsilon}) + \left(1 - \frac{n}{|\mathcal{X}| - 1} P_{\mathcal{M}^\varepsilon}\right) - 1 = 1 - \frac{n + |\mathcal{X}| - 1}{|\mathcal{X}| - 1} P_{\mathcal{M}^\varepsilon} \tag{20}$$

where (20) uses (15) and (17). Since $\mathcal{A}_{\mathsf{FC}}^{\mathcal{D}}$ can be constructed in $\mathcal{O}(d_X^2)$, we have the Theorem. $\quad\square$

### D.2. Memorization capabilities of Attention layers.

We now state Lemma 1 bounding the error of the self-attention layer in memorization mode (Vu et al., 2024). The Lemma can be considered as a specific case of Theorem 5 of (Ramsauer et al., 2021). In the context of that work, they use the term *for well-separated pattern* in their main manuscript to indicate the condition that the Theorem holds. In fact, the condition (21) stated in Lemma 1 is a sufficient condition for that Theorem of (Ramsauer et al., 2021).

**Lemma 1.** *Given a data $X$, a constant $\alpha > 0$ large enough such that, for an $x_i \in X$:*

$$\Delta_i \geq \frac{2}{\alpha N_X} + \frac{1}{\alpha} \log(2(N_X - 1) N_X \alpha M^2) \tag{21}$$

*then, for any $\xi$ such that $\|\xi - x_i\| \leq \frac{1}{\alpha N_X M}$, we have*

$$\left\| x_i - X \mathrm{softmax}\left(\alpha X^\top \xi\right)\right\| \leq 2M(N_X - 1) \exp\left(2/N_X - \alpha \Delta_i\right)$$

*Proof.* See Lemma 3 (Vu et al., 2024) for proof or Theorem 5 (Ramsauer et al., 2021) for proof of general case. $\quad\square$

Intuitively, Lemma 1 claims that, if we have a pattern $\xi$ near $x_i$, $X\mathrm{softmax}\left(\alpha X^\top \xi\right)$ is exponentially near $\xi$ as a function of $\Delta_i$. Another key remark of the Lemma is that $\left\|x_i - X\mathrm{softmax}\left(\alpha X^\top \xi\right)\right\|$ exponentially approaches 0 as the input dimension increases (Ramsauer et al., 2021).

We now consider the iteration $\xi^{\mathrm{new}} = f(\xi) = Xp = X\,\mathrm{softmax}(\beta X^T \xi)$. We now state Lemma 2 (Lemma A3 (Ramsauer et al., 2021)) that provide the bound on the Jacobian of the fixed point iteration:

**Lemma 2.** *The following bound on the norm $\|J\|_2$ of the Jacobian of the fixed point iteration $f$ holds independent of $p$ or the query $\xi$:*

$$\|J\|_2 \leq \beta m_{max}^2, \tag{22}$$

*Proof.* See Lemma A3 (Ramsauer et al., 2021). $\quad\square$

## D.3. The Separation of LDP-Protected Data

To give an intuition that the LDP mechanism generally increases the separation of the data, we consider the LDP mechanism in which each $r_i$ is i.i.d sampled and independently added to each input pattern. We have the expected value of the separation between two patterns is:

$$\mathbf{E}\left[x_i^{\varepsilon\top}x_i^{\varepsilon} - x_i^{\varepsilon\top}x_j^{\varepsilon}\right] = \mathbf{E}\left[(x_i + r_i)^{\top}(x_i + r_i) - (x_i + r_i)^{\top}(x_j + r_j)\right] \tag{23}$$

$$=\mathbf{E}\left[x_i^{\top}x_i - x_i^{\top}x_j\right] + 2\mathbf{E}\left[x_i^{\top}r_i\right] - \mathbf{E}\left[r_i^{\top}x_j\right] - \mathbf{E}\left[r_j^{\top}x_i\right] + \mathbf{E}\left[r_i^{\top}r_i\right] - \mathbf{E}\left[r_i^{\top}r_j\right] \tag{24}$$

$$=\mathbf{E}\left[x_i^{\top}x_i - x_i^{\top}x_j\right] + \mathrm{Var}(r_i) \tag{25}$$

As we can see, the last expression is the expectation of the separation of the data $D$ plus the variance of the noise. Thus, we can assume that $\Delta^{\varepsilon}$ resulting from the defense mechanism is to be at least similar to the separation $\Delta$ of the original data $D$.

## D.4. Proof of Theorem 3 on the Vulnerability of LDP-Protected Data to Attention-based AMI in FL

We are now ready to state the proof of Theorem 3. We restate the Theorem below.

**Theorem.** *Given a $\Delta^{\varepsilon}$-separated data $D^{\mathcal{M}_{\varepsilon}}$ (the LDP-protected version of the data D) with i.i.d patterns of the security game $\mathsf{Exp}_{\mathrm{LDP}}^{\mathrm{AMI}}$, for any $\beta > 0$ large enough such that:*

$$\Delta^{\varepsilon} \geq \frac{2}{\beta N_X} + \frac{1}{\beta}\log(2(N_X - 1)N_X\beta M^{\varepsilon 2}) \tag{26}$$

*then there exists an AMI adversary that exploits the self-attention layer $\mathcal{A}_{\mathrm{Attn}}^{\mathcal{D}}$ whose complexity is $\mathcal{O}(d_X^3)$ of the threat model $\mathsf{Exp}_{\mathrm{LDP}}^{\mathrm{AMI}}$ such that $\mathbf{Adv}_{\mathrm{LDP}}^{\mathrm{AMI}}(\mathcal{A}_{\mathrm{Attn}}^{\mathcal{D}})$ is lower bounded by:*

$$\mathbf{Adv}_{\mathrm{LDP}}^{\mathrm{AMI}}(\mathcal{A}_{\mathrm{Attn}}^{\mathcal{D}}) \geq P_{\mathrm{proj}}^{\mathcal{D}^{\mathcal{M}_{\varepsilon}}}\left(\frac{1}{\beta N_X M^{\varepsilon}}\right) + P_{\mathrm{proj}}^{\mathcal{D}^{\mathcal{M}_{\varepsilon}}}\left(\frac{1}{\beta N_X M^{\varepsilon}}\right)^{2nN_X} - P_{\mathrm{box}}^{\mathcal{D}^{\mathcal{M}_{\varepsilon}}}\left(3\bar{\Delta}^{\varepsilon} + \beta(m_{max}^{\varepsilon})^2 R^{\varepsilon}\right) - 1 \tag{27}$$

*where $\bar{\Delta}^{\varepsilon} := 2M^{\varepsilon}(N_X - 1)\exp\left(2/N_X - \beta\Delta^{\varepsilon}\right)$ and $\mathcal{D}^{\mathcal{M}_{\varepsilon}}$ is the distribution of the protected data $D^{\mathcal{M}_{\varepsilon}}$ induced by the original data distribution $\mathcal{D}$ and the LDP-mechanism $\mathcal{M}_{\varepsilon}$. $m_x^{\varepsilon} = \frac{1}{N_X}\sum_{i=1}^{N_X}x_i^{\varepsilon}$ is the arithmetic mean of all LDP-protected patterns and $m_{max}^{\varepsilon} = \max_{1 \leq i \leq N_X}\|x_i - m_x^{\varepsilon}\|$. Here, $P_{\mathrm{proj}}^{\mathcal{D}^{\mathcal{M}_{\varepsilon}}}(\delta)$ is the probability that the projected component between two independent patterns drawn from $\mathcal{D}^{\mathcal{M}_{\varepsilon}}$ is smaller than $\delta$ and $P_{\mathrm{box}}^{\mathcal{D}^{\mathcal{M}_{\varepsilon}}}(\delta)$ is the probability that a random pattern drawn from $\mathcal{D}^{\mathcal{M}_{\varepsilon}}$ is in the cube of size $2\delta$ centering at the arithmetic mean of the patterns in $\mathcal{D}^{\mathcal{M}_{\varepsilon}}$.*

*Proof.* We represent the victim's dataset as $D = \{X_i\}_{i=1}^{n}$, where $X_i \in \mathcal{X}$, and $\mathcal{X} \subseteq \mathbb{R}^{d_X \times N_X}$. For any 2-dimensional array $X$, each column $x_j \in \mathbb{R}^{d_X}$ is referred to as a *pattern*. The distortion imposed by LDP is modeled by a noise $r_i$ added to each pattern: $X^{\varepsilon} = \mathcal{M}^{\varepsilon}(X) = \{x_i + r_i\}_{i=1}^{N_X} = \{x_i^{\varepsilon}\}_{i=1}^{N_X}$. For brevity, we first consider the following notation of the output of one attention head under LDP without the head indexing $h$:

$$X^{\varepsilon}\mathrm{softmax}\left(1/\sqrt{d_{\mathrm{attn}}}(X^{\varepsilon})^{\top}W_K^{\top}W_Q X^{\varepsilon}\right) \tag{28}$$

Notice that we omit $W_V$ because they are all set to identity ( line 14 Algo. 3).

To show the Theorem, we consider the AMI adversary $\mathcal{A}_{\mathrm{Attn}}^{\mathcal{D}}$ specified in Subsection 4.2. Since $W \in \mathbb{R}^{d_X \times d_X}$ (line 5 in Algorithm 3) is initialized randomly, it has a high probability of being non-singular, even after assigning $v$ to its first column (line 6 in Algorithm 3). For simplicity of analysis, we assume that $W$ has full rank. If this assumption does not hold, we can re-run the two corresponding lines of the algorithm. Similarly, we also assume that all $W_Q^h$ and $W_K^h$ have rank $d_{\mathrm{attn}} = d_X - 1$.

For all heads, we set $W_K = \beta(W_Q^{\top})^{\dagger}$ (lines 9 and 10 in Algorithm 3). Consequently, $\frac{1}{\beta}W_K^{\top}W_Q = W_Q^{\dagger}W_Q$ is the projection matrix onto the column space of $W_Q^{\top}$. By defining $[\xi_1, \cdots, \xi_{N_x}] = \Xi = \frac{1}{\beta}W_K^{\top}W_Q X^{\varepsilon}$, it follows that $\xi_j$ is the projection of the pattern $x_j^{\varepsilon}$ onto this space.

For head 1 and head 3, as a result of line 6 in Algorithm 3, we can express $W = [v, w_2, \cdots, w_{d_X}]$. Based on the QR factorization (line 7), we have:

$$QR = [v, w_2, \cdots w_{d_X}] \longrightarrow R = Q^{\top}[v, w_2, \cdots w_{d_X}] \tag{29}$$

Since $R$ is an upper triangular matrix, $v$ is orthogonal to all rows $Q_i$ for $i \in \{2, \cdots, d_X\}$ of $Q^\top$. Additionally, due to the assignment at line 8 in Algorithm 3, the column space of $W_Q^\top$ is the linear span of $\{Q_i\}_{i=2}^{d_X}$, all of which are orthogonal to $v$. As a result, the difference between $X^\varepsilon$ and $\Xi$ corresponds to the component of $X^\varepsilon$ in the direction of $v$:

$$X^\varepsilon - \Xi^h = [x_1^\varepsilon - \xi_1^h, \cdots, x_{N_X}^\varepsilon - \xi_{N_X}^h] = [\mathrm{Proj}_v(x_1^\varepsilon), \cdots, \mathrm{Proj}_v(x_{N_X}^\varepsilon)], \quad h \in \{1, 3\} \tag{30}$$

where $\mathrm{Proj}_v(x_j^\varepsilon)$ is the component of pattern $x_j^\varepsilon \in \mathbb{R}^{d_X}$ along $v$.

For head 2 and head 4, although QR-factorization is not performed, $\frac{1}{\beta} W_K^\top W_Q$ for these heads also acts as projection matrices, but onto different column spaces. These spaces are likewise of rank $d_X - 1$, and each omits one direction. Denoting this direction as $u$, we can express the difference between $X^\varepsilon$ and $\Xi$ for these heads as:

$$X^\varepsilon - \Xi^h = [x_1^\varepsilon - \xi_1^h, \cdots, x_{N_X}^\varepsilon - \xi_{N_X}^h] = [\mathrm{Proj}_u(x_1^\varepsilon), \cdots, \mathrm{Proj}_u(x_{N_X}^\varepsilon)], \quad h \in \{2, 4\} \tag{31}$$

We now denote $f_\alpha : \mathbb{R}^{d_X \times N_x} \to \mathbb{R}^{d_X \times N_x}$ as:

$$\Xi' = f_\alpha(\Xi) = X^\varepsilon \mathrm{softmax}\left(\alpha(X^\varepsilon)^\top \Xi\right) \tag{32}$$

For brevity, we also abuse the notation and write $\xi' = f_\alpha(\xi) = X^\varepsilon \mathrm{softmax}\left(\alpha(X^\varepsilon)^\top \xi\right)$ for $\xi$ and $\xi' \in \mathbb{R}^{d_X}$.

With this, the output of the layer before the ReLU activation can be expressed as:

$$Z = \begin{bmatrix} f_\beta(\Xi^1) - f_\beta(\Xi^2) - \gamma 1^\top \\ f_\beta(\Xi^4) - f_\beta(\Xi^3) - \gamma 1^\top \end{bmatrix} = \begin{bmatrix} f_\beta(\Xi^1) - f_\beta(\Xi^2) - \gamma 1^\top \\ f_\beta(\Xi^2) - f_\beta(\Xi^1) - \gamma 1^\top \end{bmatrix} \tag{33}$$

$$= \begin{bmatrix} f_\beta(\xi_1^1) - f_\beta(\xi_1^2) - \gamma, & \cdots, & f_\beta(\xi_{N_X}^1) - f_\beta(\xi_{N_X}^2) - \gamma \\ f_\beta(\xi_1^2) - f_\beta(\xi_1^1) - \gamma, & \cdots, & f_\beta(\xi_{N_X}^2) - f_\beta(\xi_{N_X}^1) - \gamma \end{bmatrix} \tag{34}$$

where $\beta = \beta/\sqrt{d_{\text{attn}}}$ and $\gamma$ is defined as in Algorithm 3. From the above expressions, it follows that $Z$ has non-zero entries if and only if:

$$\exists i \text{ such that } \|f_\beta(\xi_i^1) - f_\beta(\xi_i^2)\|_\infty > \gamma \tag{35}$$

Note that heads 3 and 4 are used to handle cases where the entries of $f_\beta(\xi_i^1)$ are smaller than those in $f_\beta(\xi_i^2)$. The condition (35) can also be rewritten as:

$$\|f_\beta(\Xi^1) - f_\beta(\Xi^2)\|_\infty > \gamma \tag{36}$$

For a given pattern $x_i$, we now examine two cases: $x_i \neq v$ and $x_i = v$.

**Case 1.** For a pattern $x_i \in X$ such that $x_i \neq v$, from Lemma 1, we have

$$\left\| x_i^\varepsilon - f_\beta(\xi_i^1) \right\| \leq 2M^\varepsilon(N_X - 1)\exp\left(2/N_X - \beta\Delta_i^\varepsilon\right) \tag{37}$$

$$\left\| x_i^\varepsilon - f_\beta(\xi_i^2) \right\| \leq 2M^\varepsilon(N_X - 1)\exp\left(2/N_X - \beta\Delta_i^\varepsilon\right) \tag{38}$$

when $\|x_i^\varepsilon - \xi_i^1\| = \|\mathrm{Proj}_v(x_i^\varepsilon)\| \leq 1/(\beta N_X M^\varepsilon)$ and $\|\mathrm{Proj}_u(x_i^\varepsilon)\| \leq 1/(\beta N_X M^\varepsilon)$, respectively. Note that it is necessary to select a sufficiently large $\beta$ to ensure that Lemma 1 holds. We denote these events by $A_i^1$ and $A_i^2$.

Using triangle inequality, we have:

$$\|f_\beta(\xi_i^1) - f_\beta(\xi_i^2)\| \leq \left\| f_\beta(\xi_i^1) - x_i^\varepsilon \right\| + \left\| x_i^\varepsilon - f_\beta(\xi_i^2) \right\| \leq 4M^\varepsilon(N_X - 1)\exp\left(2/N_X - \beta\Delta_i^\varepsilon\right) := 2\bar{\Delta}_i^\varepsilon \tag{39}$$

with a probability of $\Pr\left[A_i^1 \cap A_i^2\right]$. Here, we define $\bar{\Delta}_i^\varepsilon := 2M^\varepsilon(N_X - 1)\exp\left(2/N_X - \beta\Delta_i^\varepsilon\right)$. We further relax the inequality using the infinity-norm, which bounds the maximum absolute difference in the pattern's feature:

$$\|f_\beta(\xi_i^1) - f_\beta(\xi_i^2)\|_\infty \leq 2\bar{\Delta}_i^\varepsilon \tag{40}$$

Since the data point $X^\varepsilon$ is $\Delta^\varepsilon$-separated, i.e., $\Delta^\varepsilon \leq \Delta_i^\varepsilon$, we further obtain:

$$\|f_\beta(\xi_i^1) - f_\beta(\xi_i^2)\|_\infty \leq 2\bar{\Delta}^\varepsilon \tag{41}$$

where $\bar{\Delta}^\varepsilon := 2M^\varepsilon(N_X - 1)\exp(2/N_X - \beta\Delta^\varepsilon)$.

We now analyze the event $A_i^1$. Essentially, this event occurs when the component of $x_i^\varepsilon$ along $v$ is smaller than a constant determined by the data distribution $\mathcal{D}$. Moreover, since both $v$ and $x_i$ are independently drawn from the distribution (as specified in the experiment $\mathsf{Exp}_{\mathrm{LDP}}^{\mathrm{AMI}}$ (Fig. 12)), and $r_i$ is the random noise introduced by the LDP mechanism, $v$ and $x_i^\varepsilon$ can be treated as two random patterns sampled from the input distribution. Consequently, the probability of $A_1^i$ corresponds to the probability that the projected component between two random patterns is less than or equal to $\frac{1}{\beta N_X M^\varepsilon}$. Formally, for an input distribution $\mathcal{D}^{\mathcal{M}_\varepsilon}$, we denote $P_{\mathrm{proj}}^{\mathcal{D}^{\mathcal{M}_\varepsilon}}(\delta)$ as the probability that the projected component between any independent patterns drawn from $\mathcal{D}^{\mathcal{M}_\varepsilon}$ is at most $\delta$. We then have:

$$\Pr\left[A_i^1\right] = \Pr\left[\|\mathrm{Proj}_v(x_i^\varepsilon)\| \le 1/(\beta N_X M^\varepsilon)\right] = P_{\mathrm{proj}}^{\mathcal{D}^{\mathcal{M}_\varepsilon}}\left(\frac{1}{\beta N_X M^\varepsilon}\right) \tag{42}$$

by definition. Similarly, for $A_i^2$, we have:

$$\Pr\left[A_i^2\right] = \Pr\left[\|\mathrm{Proj}_u(x_i^\varepsilon)\| \le 1/(\beta N_X M^\varepsilon)\right] = P_{\mathrm{proj}}^{\mathcal{D}^{\mathcal{M}_\varepsilon}}\left(\frac{1}{\beta N_X M^\varepsilon}\right) \tag{43}$$

Since $v$ and $u$ are independent, we obtain:

$$\Pr\left[A_i^1 \cap A_i^2\right] \ge P_{\mathrm{proj}}^{\mathcal{D}^{\mathcal{M}_\varepsilon}}\left(\frac{1}{\beta N_X M^\varepsilon}\right)^2 \tag{44}$$

**Case 2.** On the other hand, when $x_i = v$, we have the output of head 1 is:

$$X^\varepsilon \mathrm{softmax}\left(\beta X^{\varepsilon\top}\xi_i^1\right) \tag{45}$$

$$= X^\varepsilon \mathrm{softmax}\left(\beta X^{\varepsilon\top}\left(x_i^\varepsilon - \mathrm{Proj}_v(x_i^\varepsilon)\right)\right) \tag{46}$$

$$= X^\varepsilon \mathrm{softmax}\left(\beta X^{\varepsilon\top}\left(v + r_i - \mathrm{Proj}_v(v + r_i)\right)\right) \tag{47}$$

$$= X^\varepsilon \mathrm{softmax}\left(\beta X^{\varepsilon\top}\left(r_i - \bar{r}_i^v\right)\right) \tag{48}$$

Thus, the difference between the output of head 1 with $\bar{X}^\varepsilon := \frac{1}{N_x}\sum_{j=1}^{N_x} x_j^\varepsilon$ can be bounded by:

$$= \left\|X^\varepsilon \mathrm{softmax}\left(\beta X^{\varepsilon\top}\xi_i^1\right) - \bar{X}^\varepsilon\right\| \tag{49}$$

$$= \left\|X^\varepsilon \mathrm{softmax}\left(\beta X^{\varepsilon\top}\xi_i^1\right) - X^\varepsilon \mathrm{softmax}\left(\beta X^{\varepsilon\top}0\right)\right\| \tag{50}$$

$$= \left\|X^\varepsilon \mathrm{softmax}\left(\beta X^{\varepsilon\top}(r_i - \bar{r}_i^v)\right) - X^\varepsilon \mathrm{softmax}\left(\beta X^{\varepsilon\top}0\right)\right\| \tag{51}$$

$$\le \left\|\max_\xi \frac{\partial f_\beta(\xi)}{\partial \xi}\right\| \|r_i - \bar{r}_i^v\| \le \left\|\max_\xi \frac{\partial f_\beta(\xi)}{\partial \xi}\right\| \|r_i\| \le \beta(m_{max}^\varepsilon)^2 R^\varepsilon \tag{52}$$

where the inequalities are due to the mean value theorem (Lemma A32 (Ramsauer et al., 2021)) and from the fact that the Jacobian of $f_\beta$, i.e., $\frac{\partial f_\beta(\xi)}{\partial \xi}$, is bounded by $\beta(m_{max}^\varepsilon)^2$ (Lemma 2). Thus, from triangle inequality, we have

$$\left\|f_\beta(\xi_i^1) - f_\beta(\xi_i^2)\right\|_\infty = \left\|f_\beta(\xi_i^1) - \bar{X}^\varepsilon + \bar{X}^\varepsilon - v - r_i + v + r_i - f_\beta(\xi_i^2)\right\|_\infty \tag{53}$$

$$\ge \left\|\bar{X}^\varepsilon - v - r_i\right\|_\infty - \left\|f_\beta(\xi_i^1) - \bar{X}^\varepsilon\right\|_\infty - \left\|v + r_i - f_\beta(\xi_i^2)\right\|_\infty \tag{54}$$

$$\ge \left\|\bar{X}^\varepsilon - v - r_i\right\|_\infty - \beta(m_{max}^\varepsilon)^2 R^\varepsilon - \bar{\Delta}_i^\varepsilon \tag{55}$$

when $A_i^2$ happens (so that $\bar{\Delta}_i^\varepsilon \ge \left\|v + r_i - f_\beta(\xi_i^2)\right\|_\infty$), whose probability is $P_{\mathrm{proj}}^{\mathcal{D}^{\mathcal{M}_\varepsilon}}\left(\frac{1}{\beta N_X M^\varepsilon}\right)$.

We have the probability that $\left\|f_\beta(\xi_i^1) - f_\beta(\xi_i^2)\right\|_\infty > 2\bar{\Delta}^\varepsilon$ is bounded by:

$$\Pr\left[\left\|f_\beta(\xi_i^1) - f_\beta(\xi_i^2)\right\|_\infty > 2\bar{\Delta}^\varepsilon\right] \tag{56}$$

$$=1 - \Pr\left[\left\|f_\beta(\xi_i^1) - f_\beta(\xi_i^2)\right\|_\infty \leq 2\bar{\Delta}^\varepsilon\right] \tag{57}$$

$$=1 - \Pr\left[\left\|f_\beta(\xi_i^1) - f_\beta(\xi_i^2)\right\|_\infty \leq 2\bar{\Delta}^\varepsilon | A_i^2\right]\Pr\left[A_i^2\right]$$
$$- \Pr\left[\left\|f_\beta(\xi_i^1) - f_\beta(\xi_i^2)\right\|_\infty \leq 2\bar{\Delta}^\varepsilon | \neg A_i^2\right]\Pr\left[\neg A_i^2\right] \tag{58}$$

$$\geq 1 - \Pr\left[\left\|f_\beta(\xi_i^1) - f_\beta(\xi_i^2)\right\|_\infty \leq 2\bar{\Delta}^\varepsilon | A_i^2\right] - \Pr\left[\neg A_i^2\right] \tag{59}$$

$$\geq 1 - \Pr\left[\left\|\bar{X}^\varepsilon - v - r_i\right\|_\infty - \beta(m_{max}^\varepsilon)^2 R^\varepsilon - \bar{\Delta}^\varepsilon \leq 2\bar{\Delta}^\varepsilon | A_i^2\right] - \Pr\left[\neg A_i^2\right] \tag{60}$$

$$= 1 - \Pr\left[\left\|\bar{X}^\varepsilon - v - r_i\right\|_\infty \leq \beta(m_{max}^\varepsilon)^2 R^\varepsilon + 3\bar{\Delta}^\varepsilon\right] - \Pr\left[\neg A_i^2\right] \tag{61}$$

$$= P_{\text{proj}}^{\mathcal{D}^{\mathcal{M}_\varepsilon}}\left(\frac{1}{\beta N_X M^\varepsilon}\right) - \Pr\left[v + r_i \in \text{Box}\left(\bar{X}^\varepsilon, \beta(m_{max}^\varepsilon)^2 R^\varepsilon + 3\bar{\Delta}^\varepsilon\right)\right] \tag{62}$$

where $\text{Box}(x,\delta)$ is the cube of size $2\delta$ centering at $x$. The inequality (60) is due to (55) and (61) is from the fact that $u$ is independent from $X$ and $v$.

We now consider $\Pr\left[v + r_i \in \text{Box}\left(\bar{X}^\varepsilon, \beta(m_{max}^\varepsilon)^2 R^\varepsilon + 3\bar{\Delta}^\varepsilon\right)\right]$, which is the probability that the pattern $v + r_i$ belongs to the cube of size $2\beta(m_{max}^\varepsilon)^2 R^\varepsilon + 6\bar{\Delta}^\varepsilon$ around the sampled mean of the LDP protected patterns in $X^\varepsilon$. We denote $P_{\text{box}}^{\mathcal{D}^{\mathcal{M}_\varepsilon}}(\delta)$ is the probability that a random pattern drawn from $\mathcal{D}^{\mathcal{M}_\varepsilon}$ is in the cube of size $2\delta$ centering at the arithmetic mean of the patterns in $\mathcal{D}^{\mathcal{M}_\varepsilon}$. If the length $N_X$ of $X$ is large enough, we have the sampled mean $\bar{X}^\varepsilon$ is near the arithmetic mean of the patterns and obtain $\Pr\left[v + r_i \in \text{Box}\left(\bar{X}^\varepsilon, \beta(m_{max}^\varepsilon)^2 R^\varepsilon + 3\bar{\Delta}^\varepsilon\right)\right] \approx P_{\text{box}}^{\mathcal{D}^{\mathcal{M}_\varepsilon}}(\beta(m_{max}^\varepsilon)^2 R^\varepsilon + 3\bar{\Delta}^\varepsilon)$.

**Back to main analysis.** From the analysis of the two cases, if $v \notin X$, we have:

$$\Pr\left[\|f_\beta(\xi_i^1) - f_\beta(\xi_i^2)\|_\infty \leq 2\bar{\Delta}\right] \geq P_{\text{proj}}^{\mathcal{D}^{\mathcal{M}_\varepsilon}}\left(\frac{1}{\beta N_X M^\varepsilon}\right)^2, \quad \forall i \in \{1, \cdots, N_X\}$$

Based on the analysis of the two cases, when $v$ is not an element of $X$, it follows that:

$$\Pr\left[\|f_\beta(\Xi^1) - f_\beta(\Xi^2)\|_\infty \leq 2\bar{\Delta}\right] = \prod_{i=1}^{N_X} \Pr\left[\|f_\beta(\xi_i^1) - f_\beta(\xi_i^2)\|_\infty \leq 2\bar{\Delta}\right] \geq P_{\text{proj}}^{\mathcal{D}^{\mathcal{M}_\varepsilon}}\left(\frac{1}{\beta N_X M^\varepsilon}\right)^{2N_X}$$

Since the data points in $D$ are sampled independently, if $v$ does not appear in $D$, we obtain:

$$\Pr\left[\|f_\beta(\Xi^1) - f_\beta(\Xi^2)\|_\infty \leq 2\bar{\Delta} \text{ for all } X \in D\right] \geq P_{\text{proj}}^{\mathcal{D}^{\mathcal{M}_\varepsilon}}\left(\frac{1}{\beta N_X M^\varepsilon}\right)^{2nN_X}$$

On the other hand, if $v \in X$, we have:

$$\exists i \in \{1, \cdots, N_X\} \text{ such that } \Pr\left[\|f_\beta(\xi_i^1) - f_\beta(\xi_i^2)\|_\infty > 2\bar{\Delta}\right] \geq 1 - P_{\text{box}}^{\mathcal{D}^{\mathcal{M}_\varepsilon}}(\beta(m_{max}^\varepsilon)^2 R^\varepsilon + 3\bar{\Delta}^\varepsilon)$$

$$\Rightarrow \Pr\left[\|f_\beta(\Xi^1) - f_\beta(\Xi^2)\|_\infty > 2\bar{\Delta}\right] \geq \Pr\left[\|f_\beta(\Xi^1) - f_\beta(\Xi^2)\|_\infty \leq 2\bar{\Delta} \text{ for all } X \in D\right]$$

$$\geq P_{\text{proj}}^{\mathcal{D}^{\mathcal{M}_\varepsilon}}\left(\frac{1}{\beta N_X M^\varepsilon}\right) - P_{\text{box}}^{\mathcal{D}^{\mathcal{M}_\varepsilon}}(\beta(m_{max}^\varepsilon)^2 R^\varepsilon + 3\bar{\Delta}^\varepsilon)$$

Thus, if pattern $v$ appears in $D$, we have:

$$\Pr\left[\exists X \in D \text{ such that } \|f_\beta(\Xi^1) - f_\beta(\Xi^2)\|_\infty > 2\bar{\Delta}\right] \geq P_{\text{proj}}^{\mathcal{D}^{\mathcal{M}_\varepsilon}}\left(\frac{1}{\beta N_X M^\varepsilon}\right) - P_{\text{box}}^{\mathcal{D}^{\mathcal{M}_\varepsilon}}(\beta(m_{max}^\varepsilon)^2 R^\varepsilon + 3\bar{\Delta}^\varepsilon)$$

By choosing $\gamma = 2\bar{\Delta}^\varepsilon$, we have the probability that the adversary wins is:

$$P_W = \Pr\left[v \in D\right] \Pr\left[\|\dot{\theta}_1(W^O)\|_\infty > 0 | v \in D\right] + \Pr\left[v \notin D\right] \Pr\left[\|\dot{\theta}_1(W^O)\|_\infty = 0 | v \notin D\right] \tag{63}$$

$$= \frac{1}{2} \Pr\left[\exists X \in D \text{ such that } \|f_\beta(\Xi^1) - f_\beta(\Xi^2)\|_\infty > 2\bar{\Delta} | v \in D\right]$$

$$+ \frac{1}{2} \Pr\left[\|f_\beta(\Xi^1) - f_\beta(\Xi^2)\|_\infty \leq 2\bar{\Delta} \text{ for all } X \in D | v \notin D\right] \tag{64}$$

$$\geq \frac{1}{2} \left( P^{\mathcal{D}^{\mathcal{M}_\varepsilon}}_{\text{proj}} \left(\frac{1}{\beta N_X M^\varepsilon}\right) - P^{\mathcal{D}^{\mathcal{M}_\varepsilon}}_{\text{box}} (\beta(m^\varepsilon_{max})^2 R^\varepsilon + 3\bar{\Delta}^\varepsilon) \right) + \frac{1}{2} P^{\mathcal{D}^{\mathcal{M}_\varepsilon}}_{\text{proj}} \left(\frac{1}{\beta N_X M^\varepsilon}\right)^{2nN_X} \tag{65}$$

Thus, the advantage of the adversary $\mathcal{A}^{\mathcal{D}}_{\text{Attn}}$ in Algo. 3 can be lower bounded by:

$$\mathbf{Adv}^{\text{AMI}}_{\text{LDP}}(\mathcal{A}^{\mathcal{D}}_{\text{Attn}}) = 2P_W - 1 \geq P^{\mathcal{D}^{\mathcal{M}_\varepsilon}}_{\text{proj}} \left(\frac{1}{\beta N_X M^\varepsilon}\right) + P^{\mathcal{D}^{\mathcal{M}_\varepsilon}}_{\text{proj}} \left(\frac{1}{\beta N_X M^\varepsilon}\right)^{2nN_X} - P^{\mathcal{D}^{\mathcal{M}_\varepsilon}}_{\text{box}} (\beta(m^\varepsilon_{max})^2 R^\varepsilon + 3\bar{\Delta}^\varepsilon) - 1 \tag{66}$$

Since the complexity of the adversary $\mathcal{A}^{\mathcal{D}}_{\text{Attn}}$ is $\mathcal{O}(d^3_X)$ (determined by lines 9 and 10, Algo. 3), we have the Theorem 3. $\square$

## E. Proof of lower bound of $\mathbf{Adv}^{\text{AMI}}_{\text{GRR-LDP}}(\mathcal{A}^{\mathcal{D}}_{\text{FC}})$

**Theorem 4.** *There exists an AMI adversary $\mathcal{A}^{\mathcal{D}}_{\text{FC}}$ against data protected by Generalized Randomized Response (GRR) LDP algorithm whose time complexity is $\mathcal{O}(d^2_X)$ of the threat model $\mathsf{Exp}^{\text{AMI}}_{\text{GRR-LDP}}$ such that $\mathbf{Adv}^{\text{AMI}}_{\text{GRR-LDP}}(\mathcal{A}^{\mathcal{D}}_{\text{FC}}) \geq \frac{e^\varepsilon - n}{e^\varepsilon + |\mathcal{X}| - 1}$.*

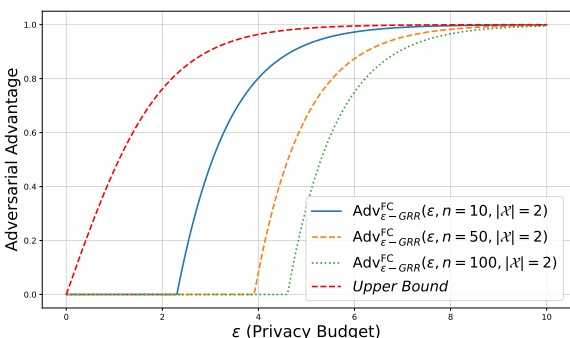

*Figure 13.* Visualization of the theoretical upperbound (Theorem. 2) and lower bound of $\mathbf{Adv}^{\text{AMI}}_{\text{GRR-LDP}}(\mathcal{A}^{\mathcal{D}}_{\text{FC}})$. (Theorem. 4)

**Generalized Randomized Response (GRR).** Given an user with a value $v \in \mathcal{X}$. A random variable, denoted by $\hat{X}$, represents the response of the user on a value $x$ also in $\mathcal{X}$. The generalized randomized response works as follows:

$$\Pr\left[\hat{X} = v\right] = \begin{cases} \frac{e^\varepsilon}{e^\varepsilon + d - 1}, & \text{if } x = v \\ \frac{1}{e^\varepsilon + d - 1}, & \text{if } x \neq v \end{cases} \tag{67}$$

where $d := |\mathcal{X}|$. Thus, we have $P_{\mathcal{M}^\varepsilon_{GRR}} = \frac{d-1}{e^\varepsilon + d - 1}$.

From (20), we have

$$\mathbf{Adv}^{\text{AMI}}_{\text{GRR-LDP}}(\mathcal{A}^{\mathcal{D}}) \geq 1 - \frac{n+d-1}{d-1} P_{\mathcal{M}^\varepsilon_{GRR}} = 1 - \frac{n+d-1}{d-1} \frac{d-1}{e^\varepsilon + d - 1} \tag{68}$$

$$= \frac{e^\varepsilon + d - 1 - n - d + 1}{e^\varepsilon + d - 1} = \frac{e^\varepsilon - n}{e^\varepsilon + d - 1} \tag{69}$$

It is clear that this advantage is smaller than the upper bound (4):

$$\frac{e^\varepsilon - n}{e^\varepsilon + d - 1} \leq \frac{e^\varepsilon - 1}{e^\varepsilon + d - 1} \leq \frac{e^\varepsilon - 1}{e^\varepsilon + 1} \tag{70}$$

# F. Proof of upper bound of $\mathbf{Adv}_{\mathbf{LDP}}^{\mathbf{AMI}}(\mathcal{A}_{\mathsf{FC}}^{\mathcal{D}})$

We now restate Theorem 2 on the theoretical upper bound of $\mathbf{Adv}_{\mathrm{LDP}}^{\mathrm{AMI}}(\mathcal{A}_{\mathsf{FC}}^{\mathcal{D}})$.

**Theorem.** *For all AMI adversary $\mathcal{A}$ of the game $\mathsf{Exp}_{\mathrm{LDP}}^{\mathsf{FC}}$, we have*

$$\mathbf{Adv}_{\mathrm{LDP}}^{\mathsf{FC}}(\mathcal{A}_{\mathsf{FC}}^{\mathcal{D}}) \leq \frac{e^{\epsilon} - 1}{e^{\epsilon} + 1}$$

*Proof.* For clarity, we denote $t_1$ the instance of $D$ that is sampled in case $b = 1$ and $t_0$ the random sample of the input distribution $\mathcal{D}$ in case $b = 0$. Let $D_1 = D$ and $D_2 = D \setminus \{t_1\} \cup \{t_0\}$. With this notations, we can think the adversary needs to differentiate $D_1$ to $D_2$ instead of $t$ or $t'$.

Using the notations as denoted in Lemma 3, the probability the adversary wins is:

$$P_W = \Pr\left[S = D_1\right] \Pr\left[\mathcal{M}_{\epsilon}(S) \in T_1^{\mathcal{M}_{\epsilon}} | S = D_1\right] + \Pr\left[S = D_2\right] \Pr\left[\mathcal{M}_{\epsilon}(S) \in T_0^{\mathcal{M}_{\epsilon}} | S = D_2\right] \tag{71}$$

$$= \frac{1}{2}\left(\Pr\left[\mathcal{M}_{\epsilon}(D_1) \in T_1^{\mathcal{M}_{\epsilon}}\right] + \Pr\left[\mathcal{M}_{\epsilon}(D_2) \in T_0^{\mathcal{M}_{\epsilon}}\right]\right) \tag{72}$$

Here, $S$ is a dummy variable representing which datasets, i.e., $D_1$ or $D_2$ is chosen by the experiment. Similarly, we have the probability the adversary loses is:

$$P_L = \frac{1}{2}\left(\Pr\left[\mathcal{M}_{\epsilon}(D_1) \in T_0^{\mathcal{M}_{\epsilon}}\right] + \Pr\left[\mathcal{M}_{\epsilon}(D_2) \in T_1^{\mathcal{M}_{\epsilon}}\right]\right) \tag{73}$$

From Lemma 3, we have:

$$\Pr\left[\mathcal{M}_{\epsilon}(D_1) \in T_1^{\mathcal{M}_{\epsilon}}\right] \leq e^{\epsilon} \Pr\left[\mathcal{M}_{\epsilon}(D_2) \in T_1^{\mathcal{M}_{\epsilon}}\right] \tag{74}$$

$$\Pr\left[\mathcal{M}_{\epsilon}(D_2) \in T_0^{\mathcal{M}_{\epsilon}}\right] \leq e^{\epsilon} \Pr\left[\mathcal{M}_{\epsilon}(D_1) \in T_0^{\mathcal{M}_{\epsilon}}\right] \tag{75}$$

Combining (72), (73), (74) and (75), we have

$$P_W = \frac{1}{2}\left(\Pr\left[\mathcal{M}_{\epsilon}(D_1) \in T_1^{\mathcal{M}_{\epsilon}}\right] + \Pr\left[\mathcal{M}_{\epsilon}(D_2) \in T_0^{\mathcal{M}_{\epsilon}}\right]\right) \tag{76}$$

$$\leq e^{\epsilon} \frac{1}{2}\left(\Pr\left[\mathcal{M}_{\epsilon}(D_1) \in T_0^{\mathcal{M}_{\epsilon}}\right] + \Pr\left[\mathcal{M}_{\epsilon}(D_2) \in T_1^{\mathcal{M}_{\epsilon}}\right]\right) = e^{\epsilon} P_L \tag{77}$$

We then have:

$$P_W(1 + e^{\epsilon}) \leq e^{\epsilon}(P_W + P_L) \Rightarrow P_W \leq e^{\epsilon}/(1 + e^{\epsilon}) \tag{78}$$

Thus, the advantage of the adversary can be bounded by:

$$\mathbf{Adv}_{\epsilon-\mathrm{DP}}^{\mathrm{AMI}}(\mathcal{A}) = 2P_W - 1 \leq \frac{e^{\epsilon} - 1}{e^{\epsilon} + 1} \tag{79}$$

We then have the Theorem. $\qquad\square$

**Lemma 3.** *Denote $\mathcal{M}_{\epsilon} : \mathcal{X} \to \mathcal{X}$ a randomized function satisfied $\epsilon$-LDP. For a database $D \in \mathcal{X}^n$, we denote $\mathcal{M}_{\epsilon}(D) := \{\mathcal{M}_{\epsilon}(x) : x \in D\}$. Then, for all database $D_1$ and $D_2$ different by one entry, we have:*

$$\Pr\left[\mathcal{M}_{\epsilon}(D_1) \in T_{b'}^{\mathcal{M}_{\epsilon}}\right] \leq e^{\epsilon} \Pr\left[\mathcal{M}_{\epsilon}(D_2) \in T_{b'}^{\mathcal{M}_{\epsilon}}\right],$$

*where $T_{b'}^{\mathcal{M}_{\epsilon}}$ is the set of all database $D$ such that the adversarial return $b'$ on that realization of $\mathcal{M}_{\epsilon}$.*

*Proof.* Let $S$ be an arbitrary subset of $\mathcal{X}$. For a pair $t, t' \in \mathcal{X} \setminus S$, consider the sets $D_1 = S \cup \{t\}$ and $D_2 = S \cup \{t'\}$. Since $\mathcal{M}_{\epsilon}$ satisfies $\epsilon$-LDP, we have:

$$\Pr\left[\mathcal{M}_{\epsilon}(t) = \mathcal{O}\right] \leq e^{\epsilon} \Pr\left[\mathcal{M}_{\epsilon}(t') = \mathcal{O}\right], \quad \forall \mathcal{O} \in \mathcal{X} \tag{80}$$

From the post-processing property of $\epsilon$-LDP, we have:

$$\Pr\left[g(\mathcal{M}_\epsilon(t)) = \mathcal{O}'\right] \leq e^\epsilon \Pr\left[g(\mathcal{M}_\epsilon(t')) = \mathcal{O}'\right], \quad \forall \mathcal{O}' \in Range(g), \tag{81}$$

for all function $g : \mathcal{X} \to Range(g)$.

In the AMI experiment, the guessing of the adversarial $\mathcal{A}$, i.e., $\mathcal{A}_{\mathsf{GUESS}}^{\mathcal{D}}(t, \dot{\theta})$, can be considered as a function on $D_{\epsilon-\mathrm{DLP}} = \mathcal{M}_\epsilon(D)$ as $\dot{\theta}$ is the result of some computations of $\mathcal{M}_\epsilon(D)$. We describe this as $\mathcal{A}(\mathcal{M}_\epsilon(D)) = b'$.

We now show that

$$\Pr\left[\mathcal{A}(\mathcal{M}_\epsilon(D_1)) = b'\right] \leq e^\epsilon \Pr\left[\mathcal{A}(\mathcal{M}_\epsilon(D_2)) = b'\right] \tag{82}$$

We show (82) by contradiction. Suppose there exists $D_1$ and $D_2$ such that the condition does not hold. We can construct a function $g : \mathcal{X} \to \{0,1\}$ as follow:

$$g(\mathcal{M}_\epsilon(x)) = \mathcal{A}\left(\mathcal{M}_\epsilon(S) \cup \{\mathcal{M}_\epsilon(x)\}\right) \tag{83}$$

With this, we have

$$\Pr\left[g(\mathcal{M}_\epsilon(t)) = b'\right] = \Pr\left[\mathcal{A}\left(\mathcal{M}_\epsilon(S) \cup \{\mathcal{M}_\epsilon(t)\}\right) = b'\right] = \Pr\left[\mathcal{A}\left(\mathcal{M}_\epsilon(D_1)\right) = b'\right] \tag{84}$$

$$> e^\epsilon \Pr\left[\mathcal{A}\left(\mathcal{M}_\epsilon(D_2)\right) = b'\right] = e^\epsilon \Pr\left[\mathcal{A}\left(\mathcal{M}_\epsilon(S) \cup \{\mathcal{M}_\epsilon(t')\}\right) = b'\right] = e^\epsilon \Pr\left[g(\mathcal{M}_\epsilon(t')) = b'\right] \tag{85}$$

which contradicts (80).

Since condition (82) is equivalent to the condition stated in the Lemma, we then have the Lemma. □

## G. Experimental Settings

This appendix outlines the experimental setup and implementation details of our work. Our experiments are implemented using Python 3.8 and executed on a single GPU-enabled compute node running a Linux 64-bit operating system. The node is allocated 36 CPU cores with 2 threads per core and 384GB of RAM. Additionally, the node is equipped with 2 RTX A6000 GPUs, each with 48GB of memory.

*Table 1.* General information of our reported experiments in the main manuscript.

| Experiments | No. runs | Adversary | Hyper-parameters | Dataset | Embedding | LDP Mechanism |
|---|---|---|---|---|---|---|
| Fig. 5, 6 | 1000 | $\mathcal{A}_{\mathsf{Attn}}^{\mathcal{D}}$ | $\beta, \gamma$ | One-hot / Spherical | No | - |
| Fig. 7,8 | $20 \times 200$ | $\mathcal{A}_{\mathsf{FC}}^{\mathcal{D}}$ | $\tau^{\mathcal{D}}, \epsilon$ | CIFAR10 / CIFAR100 | ResNet | BitRand/ GRR/ RAPPOR/ dBitFlipPM |
| Fig. 9 a,b | $20 \times 200$ | $\mathcal{A}_{\mathsf{Attn}}^{\mathcal{D}}$ | $\beta, \gamma, \epsilon$ | CIFAR10 | ViT-Base | BitRand/ GRR/ RAPPOR/ dBitFlipPM |
| Fig. 9 c | $40 \times 100$ | $\mathcal{A}_{\mathsf{Attn}}^{\mathcal{D}}$ | $\beta, \gamma, \epsilon$ | ImageNet | ViT-Base | BitRand/ GRR/ RAPPOR/ dBitFlipPM |

Table 1 shows the general information of our experiments reported in the main manuscripts. The hyper-parameters column refers to those of the adversaries, and $\epsilon$ refers to the privacy budget. The embedding specifies how the dataset is transformed to obtain the data $D$ for our testing of inference attacks. The No. runs indicates the total number of simulated security games for each point plotted in our figures. For instance, the number $20 \times 200$ means we conduct 20 trials of the experiment. Each trial consists of 200 games. The attack success rate is measured as $\frac{1}{2}(\Pr[b' = 1 | b = 1] + \Pr[b' = 0 | b = 0])$. For each trial, everything is reset. In each trial, only the LDP mechanism is re-run. Using equation 1 as well as lower (Theorem 1) and upper (Theorem 2) bound of $\mathbf{Adv}_{\mathrm{LDP}}^{\mathrm{AMI}}(\mathcal{A}_{\mathsf{FC}}^{\mathcal{D}})$, we can derive the theoretical lower and upper bound of the attack success rate.

The reported model accuracies (black lines) in all of our figures are obtained by evaluating the model on classification tasks. Particularly, for $\mathcal{A}_{\mathsf{FC}}^{\mathcal{D}}$, we use ResNet's embedding combined with a Multilayer Perceptron to generate classifications. For $\mathcal{A}_{\mathsf{FC}}^{\mathcal{D}}$, we use the native classification tasks and the original models published along with the datasets. To implement LDP, we add noise directly to ResNet's embedding. For ViTs, we add noise to the patch embeddings of the image. Since no pre-train model for ViT-B-32-224 on CIFAR10/CIFAR100 are available, we fine-tune the model that was pre-trained on ImageNet-21k on CIFAR10. The labels for classification in CIFAR10 are from the original dataset.

In the following appendices, we discuss more on the dataset and their embedding, the implementation of the adversaries, and the implementation of the LDP mechanisms. Those contents are in Appx. G.1, G.2, G.3 and G.4, respectively.

## G.1. Dataset and embedding

In total, our experiments are conducted on 2 synthetic datasets, 3 real-world datasets and 4 LDP mechanisms. The synthetic datasets are one-hot encoded data and Spherical data (data on the boundary of a unit ball). The real-world datasets are CIFAR10 (Krizhevsky et al., 2009) and ImageNet (Krizhevsky et al., 2012). The real-world datasets are pre-processed with practical pre-trained embedding modules to obtain the data $D$ in the threat models. For ResNet, we use Img2Vec (Safka, 2021) to extract the feature embeddings of images and for ViTs, we use pretrained foundation models published on HuggingFace by the authors. The parameters of $D$ in each experiment are given in Table 2.

For the synthetic datasets, we use a batch size of 1 since it does not affect the results of one-hot encoding. Furthermore, the setting also provides better intuition on the asymptotic behaviors of other datasets in Fig. 5 and Fig. 6. For ViTs, the number of patterns $N_X$ is equal to the number of image patches. Details on how $\mathcal{A}_{\text{Attn}}^{\mathcal{D}}$ works on ViTs are given in G.2. The embedding dimensions $d_X$ are determined by the choice of the embedding modules.

Table 2. Information of the data $D$ in each testing dataset.

| Dataset | Embedding | Tested batch dimension $n \times d_X \times N_X$ |
|---|---|---|
| One-hot | N/A | $1 \times [10, \cdots, 1000] \times [5, 10, 15]$ |
| Spherical | N/A | $1 \times [10000, \cdots, 35000] \times [5, 10, 15]$ |
| CIFAR10 | ResNet-18 | $64 \times 512 \times 1$ |
| CIFAR100 | ResNet-18 | $64 \times 512 \times 1$ |
| CIFAR10 | ViT-B-32-224 | $[10, 20, 40] \times 768 \times 49$ |
| ImageNet | ViT-B-32-384 | $[10, 20, 40] \times 768 \times 144$ |

## G.2. Implementation of $\mathcal{A}_{\text{Attn}}^{\mathcal{D}}$ on Vision Transformer.

First we describe the architecture of ViT, which was first proposed in (Dosovitskiy et al., 2021). First, the image is divided into $L$ fixed-size patches (e.g., $16 \times 16$ pixels or $32 \times 32$ pixels), which are flattened into vectors. Each patch is projected into a lower-dimensional embedding using a linear layer. Position embeddings are added to retain spatial information, and a learnable [class] embedding is included for global context. The sequence of embeddings is processed by $L$ Transformer encoder layers. The output corresponding to the [class] embedding is passed through an MLP head to predict the image class. In summary. ViT treats image patches like words in a sentence, using the Transformer architecture to model relationships between patches and perform tasks like image classification. For naming scheme, ViT-B-32-224 means the model is ViT-Base, the image size is 224 and the patch size is 32. Figure 14 describes ViT's architecture.

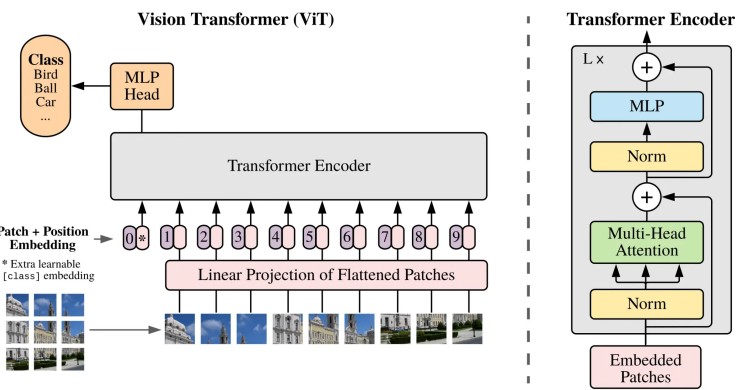

Figure 14. Architecture of Vision Transformer model. Image adapted from (Dosovitskiy et al., 2021).

Recall we represent the victim's dataset as $D = \{X_i\}_{i=1}^n$, where $X_i \in \mathcal{X}$, and $\mathcal{X} \subseteq \mathbb{R}^{d_X \times N_X}$. For any 2-dimensional array

$X$, each column $x_j \in \mathbb{R}^{d_X}$ is referred to as a *pattern*. In the context of $\mathcal{A}_{\text{Attn}}^{\mathcal{D}}$ on ViT, since $\mathcal{A}_{\text{Attn}}^{\mathcal{D}}$ operates on the pattern level, it operates directly on the Patch+Position Embedding vectors. Given the image is divided into $L$ patches, $N_X = L$ in the attack, and $d_x$ the dimension of each embedding vector. When we add LDP noise to the data, we add it directly to these vectors as well. For example, in the case of ViT-B-32-384, there are in total $\frac{384}{32} \times \frac{384}{32} = 12 \times 12 = 144$ image patches, corresponding to $N_X = 144$.

### G.3. Implementation of the Adversaries

In our theoretical analysis of $\mathcal{A}_{\text{FC}}^{\mathcal{D}}$ and $\mathcal{A}_{\text{Attn}}^{\mathcal{D}}$, we have specified how their hyper-parameters should be chosen so that theoretical guarantees can be achieved. For convenience reference, we restate those setting here:

- For Theorem 1, $\tau^{\mathcal{D}}$ is set to $\Delta^{\mathcal{X}}$. The argument is made at Subsect. 4.1.

- For Theorem 3, $\beta$ is chosen such that condition of the Theorem holds and $\gamma$ is set to $2\bar{\Delta}^{\varepsilon}$. The argument is made at Appx. D.4.

*Table 3.* Note on the values of $\beta$ in experiments.

| **Dataset** | **Note on $\beta$** | **Min $\beta$** | **Max $\beta$** |
|---|---|---|---|
| One-hot / Spherical | $\beta$ is is set to a constant | 10 | 10 |
| CIFAR10/100 | The more noise, the smaller $\beta$ | 0.01 | 0.07 |
| ImageNet | The more noise, the smaller $\beta$ | 0.01 | 0.07 |

However, as the adversaries generally know the data distribution and the LDP mechanism in practice, they can simulate the data as well as its protected version. We integrate these simulations into our implementations of $\mathcal{A}_{\text{FC}}^{\mathcal{D}}$ and $\mathcal{A}_{\text{Attn}}^{\mathcal{D}}$ to tune $\tau^{\mathcal{D}}$ and $\gamma$ before the security games. In fact, for a given LDP mechanism and an $\varepsilon$ privacy budget, the server uses a dataset from the data distribution $\mathcal{D}$ and collects the layers' outputs before the ReLU activation. Then, a linear regression model is fitted on those outputs to estimate the biases $\tau^{\mathcal{D}}$ and $\gamma$ that will be used in the security games. Regarding the hyper-parameter $\beta$, while it cannot be tuned with linear regression, for each privacy budget of a security game, we try several values of $\beta$ based on the statistic of $\mathcal{D}$ before the game and settle with a look-up table for it. An example of possible values of $\beta$ are given in Table 3. However, we found that simply setting $\beta = 0.01$ usually give satisfactory results. We use $\beta = 0.01$ for all NLP experiments.

### G.4. Details of LDP mechanisms

**Generalized Randomized Response (GRR).** Given an user with a value $v \in \mathcal{X}$. A random variable, denoted by $\hat{X}$, represents the response of the user on a value $x$ also in $\mathcal{X}$. The generalized randomized response works as follows:

$$\Pr\left[\hat{X} = v\right] = \begin{cases} \frac{e^{\varepsilon}}{e^{\varepsilon}+d-1}, & \text{if } x = v \\ \frac{1}{e^{\varepsilon}+d-1}, & \text{if } x \neq v \end{cases} \tag{86}$$

where $d := |\mathcal{X}|$.

**RAPPOR (Randomized Aggregatable Privacy-Preserving Ordinal Response).** Each user has a value $v$ encoded as a Bloom filter vector $B \in \{0,1\}^k$ using $h$ hash functions. A permanent randomized response $B'$ is generated as:

$$B_i' = \begin{cases} 1 & \text{with prob. } \frac{1}{2}f \quad \text{(flip to 1)} \\ 0 & \text{with prob. } \frac{1}{2}f \quad \text{(flip to 0)} \\ B_i & \text{with prob. } 1-f \quad \text{(keep original bit)} \end{cases}$$

Then, the instantaneous randomized response $S \in \{0,1\}^k$ is sampled from $B'$ as:

$$\Pr[S_i = 1] = \begin{cases} q & \text{if } B_i' = 1 \\ p & \text{if } B_i' = 0 \end{cases}$$

**dBitFlipPM.** Given a user with a value $v \in [k]$, the mechanism proceeds as follows:

- The user selects $d$ random buckets $\{j_1, \ldots, j_d\} \subset [k]$ without replacement.

- For each selected $j_p$, the user responds with a bit $b_{j_p} \in \{0, 1\}$ such that:

$$\Pr[b_{j_p} = 1] = \begin{cases} \frac{e^{\varepsilon/2}}{e^{\varepsilon/2}+1} & \text{if } v = j_p \\ \frac{1}{e^{\varepsilon/2}+1} & \text{if } v \neq j_p \end{cases}$$

The data collector reconstructs the histogram using:

$$\hat{h}_t(v) = \frac{k}{nd} \sum_{i=1}^{n} b_{i,v}(t) \cdot \frac{e^{\varepsilon/2}+1}{e^{\varepsilon/2}-1} - \frac{1}{e^{\varepsilon/2}-1}$$

This mechanism guarantees $\varepsilon$-LDP with reduced communication cost and supports memoization for repeated collection.

In bit-flipping mechanisms like **OME** and **BitRand**, the original data or embedding features are first converted into binary vectors. The LDP mechanisms are then applied on top of those binary representations of the signal. After that, the protected binary signals are converted back to the original domain of the data before the training of the machine learning models.

In OME, each bit $i$ of the binary representation is randomized based on the following probabilities:

$$\forall i \in [0, rl-1] : P(v'_x(i) = 1) := \begin{cases} p_{1X} = \dfrac{\alpha}{1+\alpha}, & \text{if } i \in 2j, v_x(i) = 1 \\ p_{2X} = \dfrac{1}{1+\alpha^3}, & \text{if } i \in 2j+1, v_x(i) = 1 \\ q_X = \dfrac{1}{1+\alpha\exp(\frac{\varepsilon}{rl})}, & \text{if } v_x(i) = 0 \end{cases} \tag{87}$$

where $v_x(i) \in \{0, 1\}$ is the value of the bit $i$ in the binary representation, $v'_x$ is the perturbed binary vector, $\varepsilon$ is the privacy budget, and $\alpha$ is a parameter of the algorithm.

On the other hand, BitRand introduces the bit-aware term $\frac{i\%l}{l}$ to control the randomization probabilities. That bit-aware term helps the mechanism take the location of the bit into consideration for randomization. Intuitively, BitRand aims to apply less noise to bits that have more impact on the model utility. Particularly, the probabilities of perturbation are defined as:

$$\forall i \in [0, rl-1] : P(v'_x(i) = 1) = \begin{cases} p_X = \dfrac{1}{1+\alpha\exp(\frac{i\%l}{l}\varepsilon)}, & \text{if } v_x(i) = 1 \\ q_X = \dfrac{\alpha\exp(\frac{i\%l}{l}\varepsilon)}{1+\alpha\exp(\frac{i\%l}{l}\varepsilon)}, & \text{if } v_x(i) = 0 \end{cases} \tag{88}$$

## H. Additional Experiments

### H.1. Experiments using OME mechanism

We observe that models trained on OME-protected data have almost constant performance across the tested range of the privacy budget $\varepsilon$, as illustrated in Fig. 15. The same phenomenon of OME is observed and reported in multiple previous works (Arachchige et al., 2019; Lyu et al., 2020; Nguyen et al., 2023). The reason lies in the large sensitivity of the encoded binary representation in OME weakens the effect of $\epsilon$ on the randomization probabilities (Lyu et al., 2020). With the exception of ImageNet, even with high loss in model utility, AMI adversaries still achieve a high successful inference rate.

### H.2. Empirical results on NLP datasets

Table 4, 5, 6 show the average accuracies, F1, and AUCs of AMI attacks on NLP datasets under GRR, RAPPOR and dBitFlipPM, respectively. Given the same privacy budget, $\mathcal{A}_{FC}^{\mathcal{D}}$ has higher accuracy than $\mathcal{A}_{Attn}^{\mathcal{D}}$. GRR also usually performs worse as a defense mechanism compared to RAPPOR or dBitFlipM. Attention-based AMI also performed worse on LLM data compared to vision data. This could be due to the fact that it hard to bound the norm budget $R^\varepsilon$ of noise $r_i$ due to the discrete nature of LDP noise when applied to NLP domain.

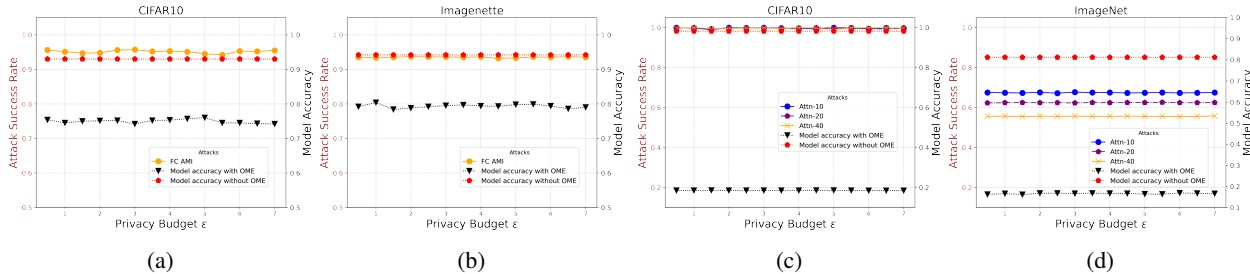

*Figure 15.* Success rates of AMI adversaries against datasets protected by OME.

*Table 4.* Average Accuracies, F1, and AUCs of AMI attacks under GRR defense.

| $\varepsilon$ | Method | BERT | | | RoBERTa | | | DistilBERT | | | GPT1 | | |
|---|---|---|---|---|---|---|---|---|---|---|---|---|---|
| | | ACC | F1 | AUC | ACC | F1 | AUC | ACC | F1 | AUC | ACC | F1 | AUC |
| $\infty$ | $\mathcal{A}_{FC}^{\mathcal{D}}$ | 1.00 | 1.00 | 1.00 | 1.00 | 1.00 | 1.00 | 1.00 | 1.00 | 1.00 | 1.00 | 1.00 | 1.00 |
| | $\mathcal{A}_{Attn}^{\mathcal{D}}$ | 1.00 | 1.00 | 1.00 | 0.96 | 0.96 | 0.99 | 1.00 | 1.00 | 1.00 | 1.00 | 1.00 | 1.00 |
| 8 | $\mathcal{A}_{FC}^{\mathcal{D}}$ | 1.00 | 1.00 | 1.00 | 1.00 | 1.00 | 1.00 | 1.00 | 1.00 | 1.00 | 1.00 | 1.00 | 1.00 |
| | $\mathcal{A}_{Attn}^{\mathcal{D}}$ | 0.99 | 0.99 | 1.00 | 0.89 | 0.88 | 0.95 | 0.99 | 0.98 | 1.00 | 0.97 | 0.97 | 0.99 |
| 6 | $\mathcal{A}_{FC}^{\mathcal{D}}$ | 0.99 | 0.99 | 1.00 | 0.97 | 0.97 | 1.00 | 0.97 | 0.97 | 1.00 | 0.99 | 0.99 | 1.00 |
| | $\mathcal{A}_{Attn}^{\mathcal{D}}$ | 0.86 | 0.84 | 0.94 | 0.75 | 0.72 | 0.82 | 0.86 | 0.83 | 0.94 | 0.86 | 0.84 | 0.93 |
| 4 | $\mathcal{A}_{FC}^{\mathcal{D}}$ | 0.86 | 0.83 | 0.91 | 0.86 | 0.85 | 0.90 | 0.83 | 0.80 | 0.87 | 0.78 | 0.75 | 0.82 |
| | $\mathcal{A}_{Attn}^{\mathcal{D}}$ | 0.56 | 0.52 | 0.59 | 0.56 | 0.53 | 0.55 | 0.57 | 0.52 | 0.60 | 0.57 | 0.53 | 0.60 |
| 2 | $\mathcal{A}_{FC}^{\mathcal{D}}$ | 0.59 | 0.57 | 0.63 | 0.59 | 0.55 | 0.62 | 0.50 | 0.50 | 0.50 | 0.55 | 0.52 | 0.52 |
| | $\mathcal{A}_{Attn}^{\mathcal{D}}$ | 0.55 | 0.50 | 0.54 | 0.50 | 0.50 | 0.50 | 0.51 | 0.50 | 0.50 | 0.50 | 0.50 | 0.52 |

*Table 5.* Average Accuracies, F1, and AUCs of AMI attacks under RAPPOR defense.

| $\varepsilon$ | Method | BERT | | | RoBERTa | | | DistilBERT | | | GPT1 | | |
|---|---|---|---|---|---|---|---|---|---|---|---|---|---|
| | | ACC | F1 | AUC | ACC | F1 | AUC | ACC | F1 | AUC | ACC | F1 | AUC |
| $\infty$ | $\mathcal{A}_{FC}^{\mathcal{D}}$ | 1.00 | 1.00 | 1.00 | 1.00 | 1.00 | 1.00 | 1.00 | 1.00 | 1.00 | 1.00 | 1.00 | 1.00 |
| | $\mathcal{A}_{Attn}^{\mathcal{D}}$ | 1.00 | 1.00 | 1.00 | 0.96 | 0.96 | 0.99 | 1.00 | 1.00 | 1.00 | 1.00 | 1.00 | 1.00 |
| 8 | $\mathcal{A}_{FC}^{\mathcal{D}}$ | 0.98 | 0.98 | 1.00 | 0.98 | 0.98 | 1.00 | 0.96 | 0.96 | 0.99 | 0.98 | 0.97 | 1.00 |
| | $\mathcal{A}_{Attn}^{\mathcal{D}}$ | 0.73 | 0.69 | 0.79 | 0.52 | 0.49 | 0.54 | 0.79 | 0.77 | 0.86 | 0.66 | 0.61 | 0.72 |
| 6 | $\mathcal{A}_{FC}^{\mathcal{D}}$ | 0.81 | 0.79 | 0.89 | 0.88 | 0.87 | 0.94 | 0.73 | 0.72 | 0.80 | 0.76 | 0.73 | 0.83 |
| | $\mathcal{A}_{Attn}^{\mathcal{D}}$ | 0.54 | 0.50 | 0.56 | 0.53 | 0.50 | 0.51 | 0.55 | 0.51 | 0.56 | 0.52 | 0.50 | 0.53 |
| 4 | $\mathcal{A}_{FC}^{\mathcal{D}}$ | 0.61 | 0.59 | 0.68 | 0.66 | 0.65 | 0.70 | 0.56 | 0.54 | 0.64 | 0.68 | 0.66 | 0.77 |
| | $\mathcal{A}_{Attn}^{\mathcal{D}}$ | 0.50 | 0.50 | 0.50 | 0.50 | 0.50 | 0.52 | 0.50 | 0.50 | 0.50 | 0.50 | 0.50 | 0.50 |
| 2 | $\mathcal{A}_{FC}^{\mathcal{D}}$ | 0.61 | 0.58 | 0.67 | 0.58 | 0.56 | 0.58 | 0.60 | 0.58 | 0.65 | 0.61 | 0.60 | 0.60 |
| | $\mathcal{A}_{Attn}^{\mathcal{D}}$ | 0.50 | 0.50 | 0.50 | 0.50 | 0.50 | 0.50 | 0.50 | 0.50 | 0.50 | 0.50 | 0.50 | 0.50 |

### H.3. ROC analysis

We also conduct an ROC analysis of the attack success rates (on IMDB dataset). GRR shows the worst privacy with attack AUCs of 0.946 ($\varepsilon = 6$) and 1.0 ($\varepsilon = 8$), while RAPPOR and dBitFlipPM provide stronger protection—achieving near-random performance at and moderate resistance at. Zoomed-in plots show that GRR leaks sensitive signals even at low FPRs and high TPRs, while RAPPOR and dBitFlipPM maintain partial robustness in these critical regions. Details are given in Fig. 16.

## I. Alternative privacy-preserving techniques beyond LDP

Other works have pursued cryptography-based techniques such as secure multi-party computation (SMPC) or homomorphic encryption (HE) that preserve privacy without adding excessive noise, hence preserving utility. SMPC-based FL systems

*Table 6.* Average Accuracies, F1, and AUCs of AMI attacks under dBitFlipPM defense.

| $\varepsilon$ | Method | BERT ACC | F1 | AUC | RoBERTa ACC | F1 | AUC | DistilBERT ACC | F1 | AUC | GPT1 ACC | F1 | AUC |
|---|---|---|---|---|---|---|---|---|---|---|---|---|---|
| $\infty$ | $\mathcal{A}_{FC}^{\mathcal{D}}$ | 1.00 | 1.00 | 1.00 | 1.00 | 1.00 | 1.00 | 1.00 | 1.00 | 1.00 | 1.00 | 1.00 | 1.00 |
| | $\mathcal{A}_{Attn}^{\mathcal{D}}$ | 1.00 | 1.00 | 1.00 | 0.96 | 0.96 | 0.99 | 1.00 | 1.00 | 1.00 | 1.00 | 1.00 | 1.00 |
| 8 | $\mathcal{A}_{FC}^{\mathcal{D}}$ | 0.96 | 0.96 | 0.99 | 0.98 | 0.98 | 1.00 | 0.96 | 0.95 | 1.00 | 0.98 | 0.97 | 0.99 |
| | $\mathcal{A}_{Attn}^{\mathcal{D}}$ | 0.76 | 0.72 | 0.84 | 0.65 | 0.59 | 0.73 | 0.76 | 0.72 | 0.84 | 0.65 | 0.59 | 0.73 |
| 6 | $\mathcal{A}_{FC}^{\mathcal{D}}$ | 0.79 | 0.77 | 0.87 | 0.84 | 0.82 | 0.92 | 0.69 | 0.67 | 0.76 | 0.77 | 0.75 | 0.82 |
| | $\mathcal{A}_{Attn}^{\mathcal{D}}$ | 0.55 | 0.50 | 0.55 | 0.53 | 0.50 | 0.51 | 0.55 | 0.50 | 0.59 | 0.52 | 0.50 | 0.53 |
| 4 | $\mathcal{A}_{FC}^{\mathcal{D}}$ | 0.65 | 0.65 | 0.74 | 0.69 | 0.68 | 0.75 | 0.50 | 0.50 | 0.51 | 0.65 | 0.63 | 0.68 |
| | $\mathcal{A}_{Attn}^{\mathcal{D}}$ | 0.50 | 0.50 | 0.50 | 0.50 | 0.50 | 0.52 | 0.50 | 0.50 | 0.50 | 0.50 | 0.50 | 0.50 |
| 2 | $\mathcal{A}_{FC}^{\mathcal{D}}$ | 0.61 | 0.59 | 0.64 | 0.56 | 0.53 | 0.59 | 0.59 | 0.56 | 0.59 | 0.63 | 0.61 | 0.67 |
| | $\mathcal{A}_{Attn}^{\mathcal{D}}$ | 0.50 | 0.50 | 0.50 | 0.50 | 0.50 | 0.50 | 0.50 | 0.50 | 0.50 | 0.50 | 0.50 | 0.50 |

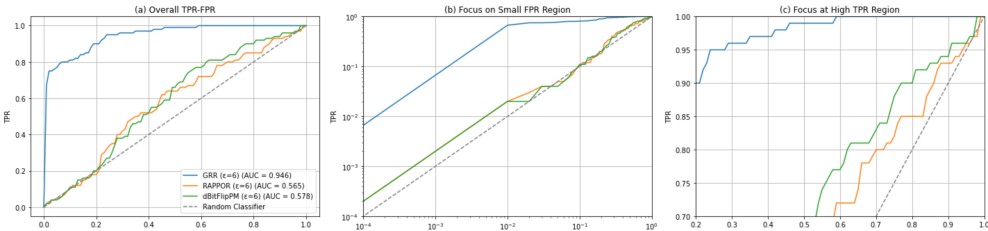

a) FPR vs TPR with AUC scores for Attention-based AMI attack against different LDP mechanisms at epsilon = 6

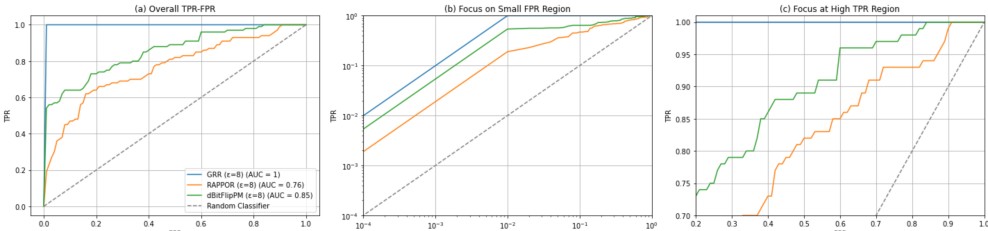

b) FPR vs TPR with AUC scores for Attention-based AMI attack against different LDP mechanisms at epsilon = 8

*Figure 16.* ROC analysis of Attention-based AMI on LDP-protected IMDB dataset.

offer strong privacy guarantees by ensuring no party learns individual updates via secure aggregation protocols (Ma et al., 2023; Bonawitz et al., 2017), but they incur high communication overhead, especially as the number of participants grows. On the other hand, HE provides end-to-end encryption in FL, allowing computations directly on encrypted data, but it introduces significant computational costs due to the complexity of cryptographic operations (Nguyen & Thai, 2023; Pan et al., 2024). Furthermore, encryption alone does not protect against inference from the final global model: if an adversary obtains the trained model, they could still perform membership inference or other attacks. Recent works combine LDP and HE/SMPC, potentially providing a comprehensive solution (Aziz et al., 2023). Additionally, we note that due to the high communication and computation overhead, secure aggregation might not be feasible in certain FL applications.

It is worth noting that secure aggregation (e.g., Secure Multi-Party Computation, Homomorphic Encryption) and Local Differential Privacy (LDP) are orthogonal research directions. SMPC/HE primarily aims to conceal local gradients from the server during aggregation, ensuring no individual gradient is exposed. LDP focuses on preventing local gradients from revealing membership information about specific data points by adding noise to the local data/ gradient.

Our threat model assumes an actively dishonest server that has access to the local gradients of clients, and conducts the proposed AMI attacks based on the local gradients. Even though using SMPC or HE could potentially conceal such info from

the server, previous research has shown that an actively dishonest adversary can circumvent secure aggregation protocols (Ngo et al., 2024; Gao et al., 2021; Liu et al., 2023; Pasquini et al., 2022; Kariyappa et al., 2023; Nguyen et al., 2022) to reconstruct the targeted client's gradients. Hence, even with secure aggregation, our threat model is still applicable when the server uses the above attacks to get around it and access local gradients before conducting the AMI attacks. Therefore, assuming that the dishonest server successfully circumvents secure aggregation, such techniques as SMPC or HE do not impact the success rates or the theoretical analysis of our proposed AMI attacks.

