# OpenReview forum: "Theoretically Unmasking Inference Attacks Against LDP-Protected Clients in Federated Vision Models"
_ICML.cc/2025/Conference — ICML 2025 poster_

### Official Review · Reviewer_exxM · 2025-03-04

**Overall Recommendation:** 3

**Summary:**

This paper studies the membership inference risk for local differential-privacy (LDP) protected clients in the presence of dishonest servers who can actively manipulate the model parameters. The paper provides theoretical upper and lower bounds for the success rates of low-polynomial-time membership attacks. It also extends a prior attack to the continuous domain of Vision Transformer models. Experiments show that the proposed attack can achieve high success rate under LDP protection.

### Update after rebuttal

I’m satisfied with the author’s response and will maintain my weak accept rating. I’d like to note that this paper is highly theoretical and falls outside my core area of expertise, so I would prefer that greater weight be given to the assessments of the other reviewers.

**Claims And Evidence:**

yes

**Essential References Not Discussed:**

-

**Experimental Designs Or Analyses:**

Yes

**Methods And Evaluation Criteria:**

yes

**Other Comments Or Suggestions:**

-

**Other Strengths And Weaknesses:**

## Strengths

1. The motivation is explained well.
2. The paper provides theoretical bounds for attacks, providing a useful tool for quantifying the risk of membership inference.
3. Experiments show that even under LDP protection, models are susceptible to membership inference attacks.

## Weaknesses
1. The problem setting used involves an active adversary who can manipulate the model weights. Extending this to include honest-but-curious adversaries would help improve the scope of the paper.

**Questions For Authors:**

-

**Relation To Broader Scientific Literature:**

This paper provides theoretical bounds to quantify the privacy risks of membership inference under an active adversary.

**Theoretical Claims:**

No

---

> ### Author Rebuttal · Authors · 2025-04-01
>
> Thank you for the positive feedback that our research is well-motivated and the provided theoretical bounds are useful for quantifying the risk of membership inference.
>
> ## #1
> `Theoretical Claims: No`
>
> We see that the reviewer indicates we do not have theoretical claims. However, we'd like to clarify that we do provide many theoretical results in the paper, as noted in your Summary section ("The paper provides theoretical upper and lower bounds for the success rates of low-polynomial-time membership attacks"). Several theoretical proofs are presented throughout the paper and in the appendix. This is also agreed on by other reviewers.
>
> ## #2
> `The problem setting used involves an active adversary who can manipulate the model weights. Extending this to include honest-but-curious adversaries would help improve the scope of the paper.`
>
> While we agree that it is interesting to see how much the assumption of having a malicious server affects the results compared to the honest-but-curious, we remain focused on the active setting due to the following reasons:
>
> - The honest-but-curious threat model assumes that the server still abides by the system protocol. This does not convey the true capability of the attacker and undermines the vulnerability of the FL system. The active adversary model is more practical because, in practice, the server can deviate from the protocol to strengthen the privacy attacks [1-4].
> - The honest-but-curious adversary model has been extensively studied and those studies achieve much lower success rates [5-7] on protected data compared to active attacks [8,9,2]. We believe this emphasizes the point that an active server introduces a much higher privacy risk, motivating the need for a more robust defense.
> - In order to tackle the more potent active threat, we intentionally focus on active adversaries (a malicious server who manipulates model weights) because this scenario represents an underexplored and more realistic threat in the federated learning literature.
>
>
> We appreciate the time and effort you have dedicated to reviewing our paper. We'd be happy to discuss further should you have any other concerns that could potentially impact your rating.
>
> ---
>
> [1] Nguyen et al. Blockchain-based secure client selection in federated learning. IEEE ICBC 2022
>
> [2] Nguyen et al. Active membership inference attack under local differential privacy in federated learning. AISTATS 2023
>
> [3] Boenisch et al. When the curious abandon honesty: Federated learning is not private. IEEE EuroS&P 2023.
>
> [4] Fowl et al. Robbing the fed: Directly obtaining private data in federated learning with modified models. ICLR 2022
>
> [5] Carlini et al. Membership inference attacks from first principles. IEEE SP 2022
>
> [6] Ye et al. Enhanced membership inference attacks against machine learning models. CCS 2022.
>
> [7] Jayaraman et al. Evaluating Differentially Private Machine Learning in Practice. USENIX Security 2019
>
> [8] Vu et al. Analysis of privacy leakage in federated large language models. AISTATS 2024
>
> [9] Nasr et al. Comprehensive privacy analysis of deep learning: Passive and active white-box inference attacks against centralized and federated learning. IEEE SP 2019.

---

### Official Review · Reviewer_x8f7 · 2025-03-08

**Overall Recommendation:** 3

**Summary:**

This paper examines the privacy risks posed by Active Membership Inference (AMI) attacks against
federated learning (FL) clients even when their data is protected by Local Differential Privacy (LDP). The
authors derive theoretical lower and upper bounds for the success rates of low-polynomial-time attacks
exploiting fully connected layers and self-attention layers and demonstrate that even under LDP, privacy
risks persist depending on the privacy budget.

**Claims And Evidence:**

Claims and evidence are sound, but the evaluation is only limited to certain types of attacks and 2 LDP
mechanisms.

**Essential References Not Discussed:**

N/A

**Experimental Designs Or Analyses:**

The experimental designs and analyses are sound, and the results align with the theoretical analysis.

**Methods And Evaluation Criteria:**

The methods and evaluation criteria are overall sound.

**Other Comments Or Suggestions:**

See Weaknesses and Questions for Authors.

**Other Strengths And Weaknesses:**

**Strengths**
1. The paper rigorously derives lower and upper bounds for the success rates of AMI attacks, providing
a mathematical basis for evaluating privacy risks under LDP.
2. The empirical results align with the theoretical analysis.

**Weaknesses**
1. The paper primarily evaluates its method under BitRand and OME, but the impact of other LDP
mechanisms remains unexplored. In particular, the study only considers noise added directly to the
data, while other common LDP approaches are perturbing the gradients before aggregation. A
discussion or experimental comparison would help clarify this aspect.
2. The paper extends the work of [1], which focuses on AMI attacks against LLMs for text data, yet it
only evaluates attacks on vision data. Given that the proposed theoretical analysis does not appear
to be domain-specific, it seems likely that it could also apply to text-based FL models. Can the authors
clarify why they chose to restrict their evaluation to image data? Providing experimental results on
NLP datasets would further strengthen the generalizability of the findings.
3. There is no Impact Statements in the paper.

[1] Vu, Minh, Truc Nguyen, and My T. Thai. &quot;Analysis of privacy leakage in federated large language
models.&quot; International Conference on Artificial Intelligence and Statistics. PMLR, 2024.

**Questions For Authors:**

1. What is the effectiveness of the proposed method for different LDP mechanisms?
2. Why do the authors only investigate vision data? Can the proposed analysis also be applied to AMIs
on NLP data since the adopted attacks are directly evaluated on text modality? Additional
experiments on NLP datasets would provide stronger evidence supporting the proposed analysis.
3. Figure 9 appears to be unclear or possibly incorrect. The right y-axis is labelled for model accuracy,
but there is no corresponding line representing the model accuracy in the figure. Can the author
provide further clarification on this issue?

**Relation To Broader Scientific Literature:**

The paper is primarily based on the analysis of [1] and [2].

[1] Vu, Minh, Truc Nguyen, and My T. Thai. &quot;Analysis of privacy leakage in federated large language
models.&quot; International Conference on Artificial Intelligence and Statistics. PMLR, 2024.

[2] Ramsauer, Hubert, et al. &quot;Hopfield networks is all you need.&quot; arXiv preprint
arXiv:2008.02217 (2020).

**Theoretical Claims:**

The overall theoretical proof in the appendix seems sound.

---

> ### Author Rebuttal · Authors · 2025-04-01
>
> ## #1
> -  `evaluation is only limited to certain types of attacks and 2 LDP mechanisms`
> - `the study only considers noise added directly to the data, while other common LDP approaches are perturbing the gradients before aggregation.`
>
> We would like to reiterate that our **theoretical** analysis applies to **all** LDP algorithms that add noise to clients' data in general, while our experiments were initially conducted using OME and BitRand. We have further conducted extra experiments using 3 other LDP algorithms, namely GRR, RAPPOR and Microsoft’s dBitFlipPM. Detailed results using these LDP algorithms are given in response #3 to Reviewer eoeX.
>
> Regarding LDP approaches that perturb gradients before aggregation, the current attack can be extended to bypass these mechanisms. First, we want to emphasize that the proposed AMI attacks hinge on the fact that the adversarial server can distinguish between non-zero gradients and zero-gradients of the targeted neurons. With gradient perturbation methods, noise (mostly Gaussian noise) is added to the clients' gradients before they are sent to the server, initially hindering the server from accessing the true value of the targeted neuron's gradient.
>
> However, previous papers [1,2] have shown that attackers can exploit knowledge of the noise distribution (e.g., Gaussian with known $\sigma$) to statistically learn the true gradient values. For instance, [1] leveraged the fact that the FL training is done in multiple iterations and the zero-mean property of Gaussian noise to average out noise samples across multiple iterations, effectively canceling the noise and revealing the true gradient values. This aligns with the central limit theorem and is consistent with the privacy composition of DP, as the privacy budget accumulates with the number of FL iterations [3]. By learning the true gradient values, the attacker can now distinguish between zero and non-zero gradients, effectively inferring whether the target sample was included in any of these iterations by applying the proposed AMI attacks.
>
> To counter such an attack, the privacy budget could be set to account for the number of iterations, however, that would result in a relatively small budget [3], making it even more difficult to have good training performance.
>
> Furthermore, theoretically analyzing membership inference attacks w.r.t adding noise to gradient has been extensively studied [4,5]. Instead, our work is the first to investigate the theoretical bound on the impact of LDP on client data.
>
> ## #2
> `can the proposed analysis also be applied to AMIs on NLP data`
>
> First, we would like to emphasize that our theoretical analysis for FC-based AMI adversary also translates to NLP (discrete) data. However, the reason why we decided to limit our theoretical analysis to vision data is due to the way we formulate the LDP noise in the Attention-based AMI adversary. In particular, the distortion imposed by LDP is modeled by a noise $r_i$ added to each pattern $x_i$ (line 247, right column). In our analysis, we assume $x_i$ and $r_i$ to be continuous, and the impact of LDP noise is visualized in Fig. 4. For NLP data, both the data and the noise should be modeled as discrete, hence our theoretical analysis might not directly apply to the NLP scenario. The key challenge is that tokens are typically represented as discrete embeddings, and adding continuous noise is not meaningful in this context.
>
> However, we note that the attacks still experimentally work against both vision and NLP data. We have conducted comprehensive experiments across 4 NLP datasets (IMDB, Yelp, Twitter, Finance), 4 models (BERT, RoBERTa, GPT-1, DistilBERT), and 3 LDP algorithms (GRR, RAPPOR, dBitFlipPM). The results indicate that privacy risks persist even for large language models (LLMs), depending on the privacy budget (https://imgur.com/a/Wt1nlno).
>
> ## #3
> `no Impact Statements in the paper`
>
> Thanks for the comment. Due to the character limit, we will include an impact statement in the revised manuscript.
>
> ## #4
> `Figure 9 appears to be unclear or possibly incorrect`
>
> We have mistakenly put that y-axis there, thanks for pointing this out. The correct figure can be found at https://imgur.com/nBpVEbB
>
> We hope our responses have addressed your concerns sufficiently, and we are happy to address any follow-up questions you might have for us.
>
> ---
>
> [1] Nguyen et al. Active membership inference attack under local differential privacy in federated learning. AISTATS 2023.
>
> [2] Hu et al. Does differential privacy really protect federated learning from gradient leakage attacks? IEEE TMC, 2024.
>
> [3] Naseri et al. Local and central differential privacy for robustness and privacy in federated learning. NDSS 2022
>
> [4] Thudi et al. From differential privacy to bounds on membership inference: Less can be more. TMLR 2024.
>
> [5] Yeom et al. Privacy risk in machine learning: Analyzing the connection to overfitting. CSF 2018.

---

### Official Review · Reviewer_eoeX · 2025-03-11

**Overall Recommendation:** 3

**Summary:**

The paper "Theoretically Unmasking Inference Attacks Against LDP-Protected Clients in Federated Vision Models" investigates the vulnerability of federated learning (FL) systems, particularly those protected by Local Differential Privacy (LDP), to Active Membership Inference (AMI) attacks. The authors derive theoretical lower bounds for the success rates of low-polynomial-time AMI attacks that exploit vulnerabilities in fully connected (FC) layers and self-attention mechanisms in vision models like ResNet and Vision Transformers (ViTs). The paper demonstrates that even with LDP protection, privacy risks persist depending on the privacy budget, and the noise required to mitigate these attacks significantly degrades model utility. The authors provide both theoretical analysis and practical evaluations, confirming that AMI attacks can achieve high success rates even under stringent LDP protection.

**Claims And Evidence:**

The claims made in the paper are supported by clear and convincing evidence. The authors provide theoretical proofs for the lower bounds of attack success rates under LDP protection (Theorems 1, 2, and 3) and validate these claims through extensive experiments on synthetic and real-world datasets (CIFAR10 and ImageNet). The experimental results align with the theoretical predictions, showing that AMI attacks can achieve high success rates even when LDP is applied, especially for smaller privacy budgets.

**Essential References Not Discussed:**

The paper adequately covers the relevant literature, but it could benefit from a discussion of recent advancements in privacy-preserving techniques beyond LDP, such as secure multi-party computation (SMPC) or homomorphic encryption, which are also used in FL systems. Additionally, the paper could discuss recent work on adversarial robustness in FL, as this is closely related to the problem of inference attacks.

**Experimental Designs Or Analyses:**

The experimental designs and analyses are sound and well-executed. The authors conduct experiments on synthetic and real-world datasets, using both FC-based and attention-based AMI attacks. The results are consistent with the theoretical predictions, showing that AMI attacks can achieve high success rates even under LDP protection, especially for smaller privacy budgets. The authors also explore the impact of hyperparameters (e.g., β) on the attack success rates, providing additional insights into the robustness of the attacks.

**Methods And Evaluation Criteria:**

The proposed methods and evaluation criteria are appropriate for the problem at hand. The authors focus on two types of AMI attacks: one exploiting FC layers and another exploiting self-attention mechanisms in transformer-based models. The evaluation is conducted on both synthetic datasets (one-hot and spherical data) and real-world datasets (CIFAR10 and ImageNet), using state-of-the-art models like ResNet and ViTs. The use of LDP mechanisms (BitRand and OME) is well-justified, and the evaluation criteria (attack success rates and model utility degradation) are relevant to assessing the trade-off between privacy and utility in FL systems.

**Other Comments Or Suggestions:**

None

**Other Strengths And Weaknesses:**

Strengths:
* The paper provides a rigorous theoretical analysis of AMI attacks under LDP protection, which is a significant contribution to the field.
* The experimental results are comprehensive and validate the theoretical claims, demonstrating the practical implications of the findings.
* The paper addresses an important gap in the literature by focusing on the vulnerabilities of LDP-protected FL systems, which are often assumed to be secure.

Weaknesses:

* The paper could benefit from a broader discussion of alternative privacy-preserving techniques beyond LDP, such as SMPC or homomorphic encryption, to provide a more comprehensive view of the privacy-utility trade-off in FL systems.
* The impact of different LDP mechanisms (e.g., BitRand vs. OME) on the attack success rates could be explored in more depth, as the current analysis focuses primarily on the theoretical bounds.

**Questions For Authors:**

1. The paper focuses on LDP as the primary privacy-preserving mechanism. Have the authors considered other privacy-preserving techniques, such as secure multi-party computation (SMPC) or homomorphic encryption, and how they might impact the success rates of AMI attacks?
2. The paper discusses the impact of the hyperparameter β on the success rates of attention-based AMI attacks. Could the authors provide more details on how

**Relation To Broader Scientific Literature:**

The paper builds on prior work in federated learning, differential privacy, and membership inference attacks. It extends the theoretical understanding of AMI attacks in FL systems, particularly under LDP protection, which has not been extensively studied in prior literature. The authors reference relevant works on LDP, FL, and AMI attacks, and their contributions are well-situated within the broader context of privacy-preserving machine learning.

**Theoretical Claims:**

The theoretical claims are well-supported by detailed proofs provided in the appendices. The authors derive lower bounds for the success rates of AMI attacks under LDP protection (Theorems 1 and 3) and an upper bound for the advantage of the adversary (Theorem 2). The proofs are rigorous and rely on established concepts in differential privacy and attention mechanisms. The theoretical analysis is a significant contribution, as it provides a formal understanding of the vulnerabilities in LDP-protected FL systems.

---

> ### Author Rebuttal · Authors · 2025-04-01
>
> ## #1
> `Have the authors considered other privacy-preserving techniques, such as secure multi-party computation (SMPC) or homomorphic encryption, and how they might impact the success rates of AMI attacks?`
>
> We thank reviewer eoeX for their insightful comments. First, we’d like to clarify that secure aggregation (e.g., Secure Multi-Party Computation, Homomorphic Encryption) and Local Differential Privacy (LDP) are orthogonal research directions. SMPC/HE primarily aims to conceal local gradients from the server during aggregation, ensuring no individual gradient is exposed. LDP focuses on preventing local gradients from revealing membership information about specific data points by adding noise to local data gradients.
>
> Our threat model assumes an actively dishonest server who has access to the local gradients of clients, and conducts the proposed AMI attacks based on the local gradients. Even though using SMPC or HE could potentially conceal such info from the server, previous research has shown that an actively dishonest adversary can circumvent secure aggregation protocols [1,2] to reconstruct the targeted client's gradients. Hence, even with secure aggregation, our threat model is still applicable when the server uses the above attacks to get around it and access local gradients before conducting the AMI attacks. Therefore, assuming that the dishonest server successfully circumvents secure aggregation, such techniques as SMPC or HE do not impact the success rates or the theoretical analysis of our proposed AMI attacks.
>
> ## #2
> `The paper could benefit from a broader discussion of alternative privacy-preserving techniques beyond LDP`
> Thanks for the suggestion. Due to the character limit in the response, we will include a discussion on secure aggregation and adversarial robustness in the revised manuscript.
>
> ## #3
> `The impact of different LDP mechanisms (e.g., BitRand vs. OME) on the attack success rates could be explored in more depth, as the current analysis focuses primarily on the theoretical bounds.`
>
> To further explore the impact of different LDP mechanisms on the attack's success rate, in addition to BitRand and OME, we have conducted extra experiments on 3 other LDP algorithms, namely GRR [3], RAPPOR [4] and Microsoft's dBitFlipPM [5]. In short, we see that privacy risks persist across all tested LDP mechanisms for both FC-based AMI and Attention-based AMI, depending on the privacy budget.
>
> We compare the attack success rate of AMI-FC across three different datasets and five distinct LDP mechanisms in this anonymized imgur link https://imgur.com/cxtqVHv. We also plot the success rate of AMI-FC w.r.t LDP mechanisms against the theoretical upper/lower bounds and privacy-utility trade-off at https://imgur.com/pZfx3Y3.
>
> For Attention-based AMI, we plot the result w.r.t. the 3 new LDP mechanisms at https://imgur.com/aDptuQq. In addition to vision datasets, we have extended our experiments to NLP datasets. The reviewer can refer to Response #2 to Reviewer x8f7 for detailed results. To explore more in depth the impact of different LDP mechanisms on the attack success rates, we also conduct an ROC analysis of the attack success rates (on IMDB dataset). The results are posted in https://imgur.com/l3Byt79. GRR shows the worst privacy with attack AUCs of 0.946 ($\epsilon=6$) and 1.0 ($\epsilon=8$), while RAPPOR and dBitFlipPM provide stronger protection—achieving near-random performance at $\epsilon=6$ and moderate resistance at $\epsilon=8$. Zoomed-in plots show that GRR leaks sensitive signals even at low FPRs and high TPRs, while RAPPOR and dBitFlipPM maintain partial robustness in these critical regions.
>
> ## #4
> `Could the authors provide more details on how`
>
> Unfortunately, this question seems to have been cut off. Could the reviewer clarify what specific information about $\beta$ they would like more details on? In the meantime, we would like to reiterate that $\beta$ controls the extent to which the attention heads memorize the target pattern. Larger $\beta$ values can negatively impact the attack's success rate (Figure 9) against LDP-protected data. Further explanations are provided in Remark 6. However, $\beta$ also needs to be sufficiently large to satisfy the condition in Equation 5.
>
> We hope our responses have addressed your questions sufficiently. We are happy to discuss further if you have follow-up questions for us.
>
> ---
>
> [1] Dario Pasquini el al. Eluding secure aggregation in federated learning via model inconsistency. CCS 2022.
>
> [2] Sanjay Kariyappa et al. Cocktail party attack: Breaking aggregation-based privacy in federated learning using independent component analysis. ICML 2023
>
> [3] Arijit Chaudhuri and Rahul Mukerjee. Randomized response: Theory and techniques. Routledge, 2020.
>
> [4] Ulfar Erlingsson, Vasyl Pihur, and Aleksandra Korolova. Rappor: Randomized aggregatable privacy-preserving ordinal response. CCS 2014
>
> [5] Bolin Ding et al. Collecting telemetry data privately. NeurIPS 2017

---

> > ### Comment · Reviewer_eoeX · 2025-04-01
> >
> > Sorry for Q3, I intended to learn about how the β was chosen in Q3. My other questions are clarified.

---

> > > ### Author Response · Authors · 2025-04-06
> > >
> > > First, we'd like to thank reviewer eoeX for engaging with our rebuttal and we are glad to have clarified all the other questions. We note that while larger $\beta$ values reduce the attack success rate, $\beta$ still needs to be sufficiently large to satisfy the condition specified in Equation (5). Given the assumption that the server has knowledge of the client's data distribution (as outlined in the AMI threat models in Section 3.1), the server can simulate the client’s data to compute a minimally sufficient value for $\beta$.
> > >
> > > When doing experiments, we found that setting $\beta$ to a reasonably small value (e.g., $0.01$) yielded consistently good results across realistic $\varepsilon$ values and datasets/LDP mechanisms. Unless stated otherwise, we select a fixed $\beta = 0.01$ in our experiments. As illustrated in Figure 9, for LDP-protected data,  $\beta=0.01$ generally achieves better attack success rates, particularly under small $\varepsilon$.
> > >
> > > We appreciate the time and effort you have contributed to reviewing our paper. We are happy with your positive evaluation that our paper made *a significant contribution to the field*, addressed *an important gap in the literature* with demonstrated *practical implications of the findings* and we hope the score reflects this.

---

### Decision · Program_Chairs · 2025-05-01

**Decision:**

Accept (poster)

**Comment:**

This paper studies Membership Inference Attacks  (MIA) in federated learning (FL) settings, regardless of the local differential privacy  (LDP) mechanism used. The main contribution is deriving lower bounds for the success rates of low-polynomial-time MIAs targeting fully connected and self-attention layers, showing that such attacks can succeed under LDP protection depending on the privacy budget. These theoretical results are supported by practical experiments on ResNet and Vision Transformers models.

Reviewers unanimously appreciated the paper’s rigorous theoretical contribution and its alignment with empirical results. The derivation of upper and lower bounds (along with the evaluation on both synthetic and real-world datasets) is appreciated by the reviewers. Multiple reviewers raise concerns about the narrow evaluation scope—particularly the focus on only two LDP mechanisms (BitRand and OME) and the absence of comparisons to other privacy-preserving approaches. In the rebuttal period, the authors included additional experiments.  There is also consensus that the experimental evaluation is limited to vision data, despite the theoretical results being generalizable to other domains such as NLP.